# GROUNDINGBOOTH: GROUNDING TEXT-TO-IMAGE CUSTOMIZATION

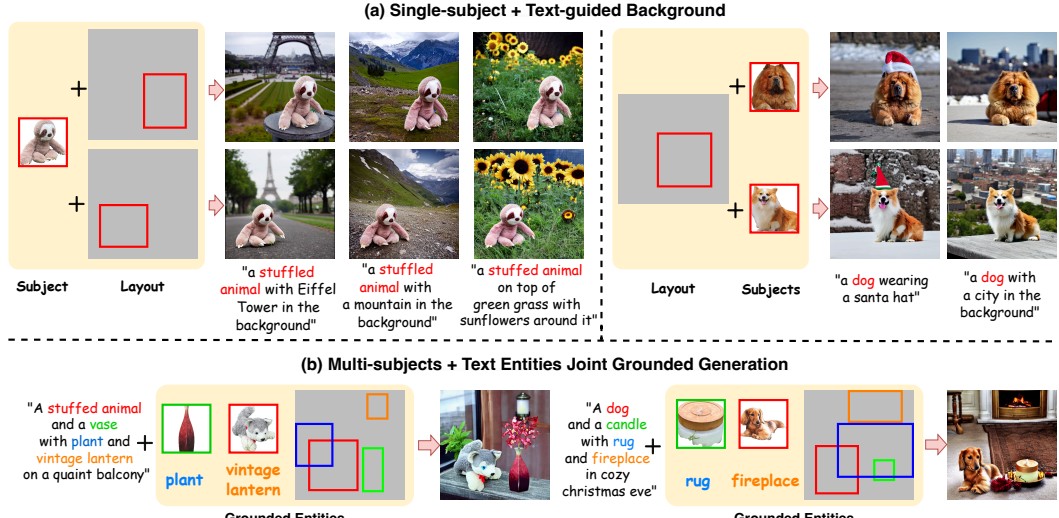

Figure 1: We propose GroundingBooth, a framework for grounded text-to-image customization. GroundingBooth supports: (a) grounded single-subject customization, and (b) joint grounded customization for multi-subjects and text entities. In general, it achieves a joint grounding on the generation of both the subject-driven foreground and the text-driven background, while preserving the identity of subjects and text-image alignment.

## ABSTRACT

Recent studies in text-to-image customization show great success in generating personalized object variants given several images of a subject. While existing methods focus more on preserving the identity of the subject, they often fall short of controlling the spatial relationship between objects. In this work, we introduce GroundingBooth, a framework that achieves zero-shot instance-level spatial grounding on both foreground subjects and background objects in the text-to-image customization task. Our proposed text-image grounding module and masked cross-attention layer allow us to generate personalized images with both accurate layout alignment and identity preservation while maintaining text-image coherence. With such layout control, our model inherently enables the customization of multiple subjects at once. Our model is evaluated on both layout-guided image synthesis and reference-based customization tasks, showing strong results compared to existing methods. Our work achieves a joint grounding on both subject-driven foreground generation and text-driven background generation. Our code will be publicly available.

## 1 INTRODUCTION

Text-to-image customization, also known as subject-driven image generation or personalized text-to-image synthesis, is a task that requires a model to generate diverse variants of a subject given a set of images of the target subject. Text-to-image customization has achieved significant progress during the

past few years, allowing for more advanced image manipulation. For example, the test-time-finetuning based methods like Dreambooth (Ruiz et al., 2023), Textual Inversion (Gal et al., 2022), and Custom Diffusion (Kumari et al., 2023) use a few images of the same object to finetune a pretrained diffusion model and generate variants of the object from input prompts. The encoder-based methods such as ELITE (Wei et al., 2023) and InstantBooth (Shi et al., 2023) eliminate test-time-finetuning by learning a generalizable image encoder and attention modules. Despite their success, existing personalization methods mainly focus on generating identity-preserved images from the input prompt and fail to accurately describe the spatial relationship of objects and backgrounds. In real-world scenarios of image customization, it is a crucial user need to achieve fine-grained and accurate layout control on each of the generated objects for more flexible image manipulation.

To tackle this issue, we propose to deal with a more challenging task, *grounded text-to-image customization*, which extends the existing text-to-image customization task by enabling grounding controllability over both the foreground subjects and background objects. Specifically, under this new setting, the inputs usually include a prompt, images of subjects, and optional bounding boxes of the subjects and background text entities. The model aims to generate text-aligned background objects and identity-preserved foreground subjects, while the spatial location of all the grounded objects and subjects are exactly aligned with the input bounding boxes. It is non-trivial to achieve all these effects simultaneously, as we are indeed handling multiple tasks together and it is challenging for the model to be generalizable to all sub-tasks.

There are a few related works to handle our new task, which, however, show significant limitations. On the one hand, although existing grounded text-to-image diffusion models such as LayoutDiffusion (Zheng et al., 2023) and GLIGEN (Li et al., 2023) have made attempts at spatial controllability, they cannot achieve identity preservation of the subjects. On the other hand, subject-driven image generation methods mainly focus on the identity preservation of the reference objects, while limited attempts have been made on layout control of either subjects or background objects. There is another line of related works (Chen et al., 2023b; Song et al., 2022; 2024) that achieve customized image composition. They can control the location of the input subject under the image composition setting but are neither able to achieve text-to-image synthesis nor control the spatial location of the background contents.

To fully address our new task, we propose GroundingBooth, a general framework for grounded text-to-image customization. Specifically, based on a pretrained text-to-image model, we build a new joint text-image grounding module that encourages both the foreground subjects and background objects to accurately follow the locations indicated by the input bounding boxes. To further enhance the identity preservation of the subjects, we propose a masked cross-attention layer in the transformer blocks of the diffusion U-Net, which helps to disentangle the subject-driven foreground generation and text-driven background generation in each block, effectively preventing the false blending of multiple visual concepts in the same location and enforcing the generation of clear subjects. As shown in Fig. 1, with such dedicated designs of the model structures, our framework not only achieves grounded text-to-image customization with a single subject (Fig. 1 (a)), but also supports multi-subject customization (Fig. 1 (b)), where users can input multiple subjects along with their bounding boxes, and our model can generate each subject in the exact target region with identity preservation and scene harmonization. Meanwhile, our model also allows for the grounding of multiple background objects (Fig. 1 (b)).

The key contributions of this work can be summarized as follows.

- We propose a general framework, GroundingBooth, that achieves grounded text-to-image customization. Specifically, our model achieves joint layout grounding of both the foreground subjects and the text-guided background, while maintaining accurate identity of the subjects. Furthermore, our model enables the customization of multiple subjects.

- We propose a novel layout-guided masked cross-attention module, which disentangles the foreground subject generation and text-driven background generation through cross-attention manipulation thus avoiding false context blending.

- Experiment results show the effectiveness of our model in text-image alignment, identity preservation, and layout alignment.

## 2 RELATED WORK

**Subject-driven Image Generation**   Subject-driven image generation, also known as personalized text-to-image generation or image customization, aims to generate target images based on customized objects and a text prompt that describes the target context (Chen et al., 2023a; Pan et al., 2024; Xiao et al., 2023; Wang et al., 2024a; Avrahami et al., 2023). In this task, the specific identity of the input reference images is defined as a subject or a concept. So far, existing image customization works can be categorized into three major types. The first type is test-time-finetuning methods (Ruiz et al., 2023; Gal et al., 2022; Kumari et al., 2023). These methods first finetune a pretrained diffusion model on a few subject images so that the model is adapted to a new identifier token representing the new concept. Then they generate new images from prompts containing the identifier. Such test-time finetuning is computationally intensive. The second type of method is encoder-based customization methods (Arar et al., 2023; Wei et al., 2023; Shi et al., 2023; Zhang et al., 2024), which eliminates the test-time finetuning by learning a generalizable diffusion model that can adapt to new subjects on a large-scale training data. The generalizable model usually contains image encoders that map the subject images into dense tokens and attention modules that integrate vision tokens with text tokens. These methods can achieve much faster image customization during inference, while rely heavily on large-scale multi-view training data, and identity preservation may not be perfect in out-of-distribution cases. The third type of methods (Roich et al., 2021; Gal et al., 2023) is a combination of the first two methods, which first learn an image encoder to extract identity tokens of the input subject and then finetune the model on the subject images for a few steps.

Note that most existing subject-driven image generation methods focus on synthesizing personalized image variants from prompts. They show quite limited performance in controlling the layout of the generated scenes and modeling the spatial relationship between objects. On the contrary, our model performs well not only in generating identity-preserved, text-aligned images but also in controlling the layout of both the subjects and background. A previous work, Break-A-Scene (Avrahami et al., 2023), employs textual scene decomposition to extract multiple text tokens from a single scene image, enabling the generation of novel images based on text prompts that feature individual concepts or combinations of multiple concepts. Both this method and our method achieve foreground-subject grounded generation. However, this method relies on test-time fine-tuning, making inference slow and computationally intensive. Additionally, there is no clear evidence in their paper that they can perform background text-entity grounding for the text-driven background objects, while our work achieves layout-guided generation of both subject-driven foreground and the text-driven background.

**Grounded Text-to-Image Generation**   Given a layout containing bounding boxes labeled with object categories, grounded text-to-image generation aims to generate the corresponding image, which is the reverse object detection process. Traditional grounded text-to-image generation such as LostGAN (Sun & Wu, 2019), LAMA (Li et al., 2021) and PLGAN (Wang et al., 2022) are based on generative adversarial networks(GANs). Recently, diffusion-based methods (Rombach et al., 2022; Zheng et al., 2023; Li et al., 2023; Zhang et al., 2023; Wang et al., 2024b) have made attempts to add layout control for image generation. For example, LayoutDiffusion (Zheng et al., 2023) uses a patch-based fusion method. GLIGEN (Li et al., 2023) injects grounded embeddings into gated Transformer layers. ControlNet (Zhang et al., 2023) uses copied encoders and zero convolutions. InstanceDiffusion (Wang et al., 2024b) allows for multiple formats of location control. LayoutGPT (Feng et al., 2024) and LayoutLLM-T2I (Qu et al., 2023) use LLM as guidance. However, all these methods can only perform text-to-image generation, while object-guided generation and identity preservation cannot be achieved. In contrast, our model achieves satisfactory identity preservation on reference-guided image generation.

## 3 OUR APPROACH

Our model is built upon Stable Diffusion v1.4 Rombach et al. (2022). Given one or a few background-free[1] reference images $\mathcal{X} = \{x_1, x_2, \cdots, x_m\}$ where $x_m \in \mathbb{R}^{h \times w \times 3}$ with their target bounding box locations $\mathcal{L}_X = \{l_X^1, l_X^2, \cdots, l_X^m\}$, text entities[2] $\mathcal{T} = \{t_1, t_2, \cdots, t_n\}$ with their target locations

---

[1]Background-free images refer to images with background removed. They can be easily obtained by segmentation methods such as SAM (Kirillov et al., 2023) or SAM2 (Ravi et al., 2024)

[2]Here each text entity is referred to a text tag, such as "chair" and "hat".

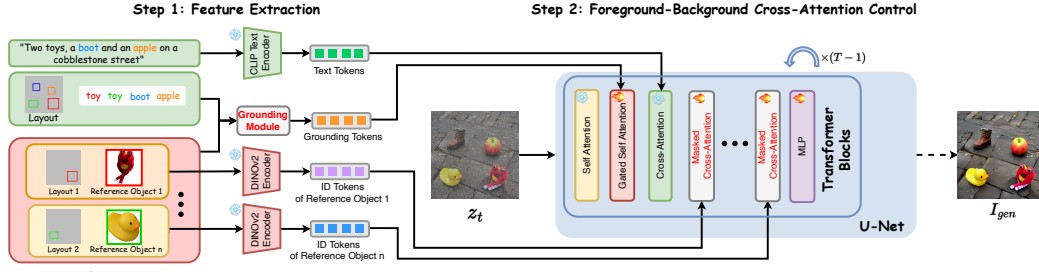

Figure 2: An overview of our GroundingBooth model. The whole pipeline contains two steps: (1) Feature extraction. We use the CLIP encoder and DINOv2 encoder to extract text and image embeddings, respectively. We use our proposed grounding module to extract the grounding tokens. (2) Foreground-background cross-attention control in each transformer block of U-Net. During training, we use datasets with a single subject as the reference image and only trains a single masked cross-attention layer per transformer block. During inference, our model supports the generation of multiple subjects in their corresponding locations by reusing the same masked cross-attention layer for each subject. This figure shows the inference pipeline of our model. We show the details of the grounding module and masked cross-attention layer in Fig. 3.

$\mathcal{L}_T = \{l_T^1, l_T^2, \cdots, l_T^n\}$, and the overall prompt $\mathcal{P}$, we aim to generate a customized image $\hat{x}$, where both the reference objects can be seamlessly placed inside the desired bounding box with natural poses and accurate identity and the background objects generated from text-box pairs are positioned at the correct location. Here $l_X^m$ or $l_T^n$ refers to the bounding box coordinates of a reference object or a text entity, which can be represented as $l = [x_{\min}, y_{\min}, x_{\max}, y_{\max}]$. The reference object should be generated harmoniously with the background. The customized image $\hat{x}$ can be calculated as:

$$\hat{x} = \text{GroundingBooth}(\mathcal{X}, \mathcal{T}, \mathcal{P}, \mathcal{L}_X, \mathcal{L}_T). \tag{1}$$

The pipeline of our proposed GroundingBooth model is shown in Fig. 2. Our work is the first attempt that enables precise spatial grounding in the customized image synthesis task, which jointly controls the size and location for both the reference-guided foreground objects and text-driven background regions. Moreover, our work adaptively harmonizes the poses of the reference objects and faithfully preserves their identity. In this section, we first introduce our feature extraction pipeline in Sec. 3.1, then introduce the foreground-background masked cross-attention control in Sec. 3.2. Finally, we introduce the training and inference pipeline in Sec. 3.3 and Sec. 3.4, respectively.

## 3.1 FEATURE EXTRACTION

**Feature Extraction of Text and Reference Images** We first extract text tokens from the input prompt using the CLIP text encoder and identity tokens from the reference images using DINOv2 (Oquab et al., 2023). For each reference image, we extract 257 identity tokens which are composed of a global image class token and 256 local patch tokens. We reshape the feature dimension of each image token to 768 through a linear projection layer.

**Grounding Module** To control the layout of the foreground and background objects, we propose a grounding module, which jointly ground text and image features through positional encoding. Fig. 3(a) shows the overall structure of our grounding module. We extract grounding information based on the joint guidance of the tagged text-box pair and the object-layout pair. Specifically, it contains two branches: 1) In the text entity branch, the bounding boxes of the background objects $\mathcal{L}_T$ are passed through a Fourier encoder to obtain the text Fourier embeddings for the text entities, which are then concatenated with the text tokens in the feature space to obtain the grounded text embeddings. 2) In the reference image branch, the bounding boxes of the reference objects $\mathcal{L}_X$ (in the target image) are also passed through a Fourier encoder to extract the object Fourier embeddings, which are then concatenated with the reference image tokens to obtain the grounded reference image embeddings. For all training images, we set a max number of boxes and the text entities and drop the rest ones. For the cases where there is no reference image or text entities, we set the input reference object layout to zero and reference object token to zero embeddings, or set the grounded text embeddings to zero

**(a) Grounding Module**     **(b) Masked Cross Attention**

Figure 3: Modules of our proposed framework: (a) Grounding Module: Our grounding module takes both the prompt-layout pairs and reference object-layout pairs as input. For the foreground reference object, both CLIP text token and the DINOv2 image class token are utilized. (b) Masked Cross-Attention: Q, K, and V are visual query, key, and value respectively, and A is the affinity matrix.

embeddings, respectively. At the end of the following two branches, the grounded text embeddings and reference image embeddings are reshaped back into the original feature dimension through linear layers and then concatenated in the token space to form the final grounding tokens. Given the text entities $\mathcal{T}$ and reference images $\mathcal{X}$, the calculation of the grounding tokens is formulated as:

$$h^{(\mathcal{T},\mathcal{X})} = \left[MLP\left(\psi_{\text{text}}\left(\mathcal{T}\right), Fourier(\mathcal{L}_T)\right), MLP\left(\psi_{\text{obj}}\left(\mathcal{X}\right), Fourier(\mathcal{L}_X)\right)\right], \quad (2)$$

where *Fourier* represents the Fourier embedding (Tancik et al., 2020), $MLP(.,.)$ is a multi-layer perceptron, $[.,.]$ is concatenation operation, and $h^{(\mathcal{T},\mathcal{X})}$ is the grounding token. $\psi_{text}$ and $\psi_{obj}$ denote to the text encoder and image encoder, respectively. The generated grounding token $h^{(\mathcal{T},\mathcal{X})}$ contains the location features of both the reference objects and the text entities. It is then injected into the U-Net layers of the diffusion models. Specifically, inspired by GLIGEN (Li et al., 2023), we inject the grounding token through a gated self-attention layer located between the self-attention layer and cross-attention layer in each Transformer block of the U-Net, represented as:

$$v = v + \tanh(\gamma) \cdot \left(\text{SelfAttn}\left(\left[v, h^{(\mathcal{T},\mathcal{X})}\right]\right)\right), \quad (3)$$

where $\gamma$ is a learnable scalar initialized as 0, $h^{(\mathcal{T},\mathcal{X})}$ is the grounding token and $v$ is the output of the self-attention layer. During training, the model adaptively learns to adjust the weight $\gamma$ of the grounding module, which ensures stable training and balances the weight between the grounding token and the visual features.

## 3.2 FOREGROUND-BACKGROUND CROSS-ATTENTION CONTROL

Previous text-to-image generation methods usually directly concatenate the text and image tokens in the cross-attention layers, leading to two drawbacks. First, the reference objects and the text-driven background objects can be blended unnaturally. Second, for the circumstances where bounding boxes belong to the same class, the model cannot distinguish whether a bounding box belongs to a reference object or text entity, resulting in the misplacement of the reference object. To solve these issues, we propose a novel masked-cross attention module to separately generate the foreground objects and background contents. Moreover, when there are multiple reference objects, our module can clearly maintain the independence of generating each foreground object. The details of our module is illustrated in Fig. 3(b).

The original cross-attention layer can be formulated as:

$$f = \text{softmax}\left(\frac{QK^T}{\sqrt{d}}\right)V, \quad (4)$$

$$Q = \phi_Q(f), K = \phi_K\left(f_c\right), V = \phi_V\left(f_c\right), \quad (5)$$

where $\sqrt{d}$ is the scaling factor that is set as the dimension of the queries and keys, $f$ denotes to the vision feature in the layer, $f_c$ refers to the embedding of the input condition. $\phi_Q, \phi_K, \phi_V$ are the linear layers to project the features into queries, keys and values, respectively.

In our masked cross-attention layer, both the DINOv2 image tokens and the layout of the reference object are taken as input. The queries $K$ and values $Q$ are calculated from the image tokens. We first compute the affinity matrix $A$ through $A = Q \cdot K$ and get $A \in \mathbb{R}^{hw \times hw}$, where $h \times w$ indicates the resolution of the feature map in the attention layer. As we have the object layout $l_{obj}$, it is straightforward to restrict the injection of image tokens only inside the region of the target bounding box. Therefore, we reshape the layout $l_{obj}$ to $h \times w$ and generate the cross attention mask, which is formulated as:

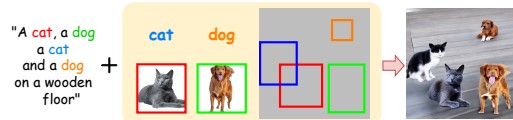

Figure 4: When the reference objects and the text entity belongs to the same class, our model can effectively prevent misplacement.

$$M_{Layout[i,j]} = \begin{cases} 0, & [i,j] \in l_{obj}, \\ -\infty, & [i,j] \notin l_{obj} \end{cases}, \tag{6}$$

where $M_{Layout[i,j]}$ represents the pixel of position $[i,j]$ in rectified attention score maps, $l_{obj}$ represents the layout region of the reference object in the feature map.

The mask contains the location information for restricting the reference object placement and avoiding information leakage. After acquiring the mask, we apply dot product operation between the feature maps and the layout to constrain the object generation and obtain the mask-rectified affinity matrix $A'$ through $A' = A + M_{Layout}$. Then we multiply the masked affine matrix $A'$ with $V$ to obtain the layout-guided masked cross-attention output $f_{obj}$. The whole masked cross-attention module is formulated as:

$$f_{obj} = \text{softmax}\left(\frac{QK^T + M_{Layout}}{\sqrt{d}}\right)V. \tag{7}$$

For the scenarios where there is a lack of reference objects, $M_{Layout}$ is set to all 0, the masked-cross attention degrade into normal cross attention. Through masked cross-attention control, the injection of each reference object feature is restricted to be within the corresponding bounding box area. This ensures not only the independence between the generation of foreground and background, but also the independence of multiple reference objects. Our module prevents information leakage and ensures an accurate layout-guided subject generation. Also, as shown in Fig. 4, when both the reference subject and the text entity belongs to the same class(cat, dog), the model can distinguish the reference object and the text entity, and effectively avoids the misplacement of the generated objects.

### 3.3 MODEL TRAINING

During training, for each image, we input only one subject image and its bounding box to the model, along with several text entities with their corresponding bounding boxes. The number of entities per training image is limited to 10 and we drop the rest ones. For a portion of training cases, the input may not contain subject image or text entities. We keep the text encoder and DINOv2 image encoder frozen and merely fine-tune the gated self-attention layers, the masked cross-attention layers, and the multi-layer perceptron after the DINOv2 image encoder.

### 3.4 MODEL INFERENCE

Although our model is trained on single-subject data, it can be seamlessly extended to achieve multi-subject customization without retraining. As shown in Fig. 2, in the inference stage, assume we have $n$ reference objects, the reference object and paired layout information are concatenated as the grounded reference image embeddings. In each transformer block, the masked cross-attention layer will be reused for $n$ times, and each ID token and its paired layout information are injected into each masked cross-attention layer respectively. As we analyzed in section 3.2, our masked cross-attention ensures the independence of the generation of each subject, preventing potential false blending of visual concepts, e.g., the unnatural blending of two objects in the overlapping regions. It also guarantees an accurate layout control on all the subjects.

### 4 EXPERIMENT

**Dataset** The training data of our experiments are from both multi-view datasets and single-image datasets. For multi-view data, we use MVImgNet (Yu et al., 2023), which contains 6.5 million frames

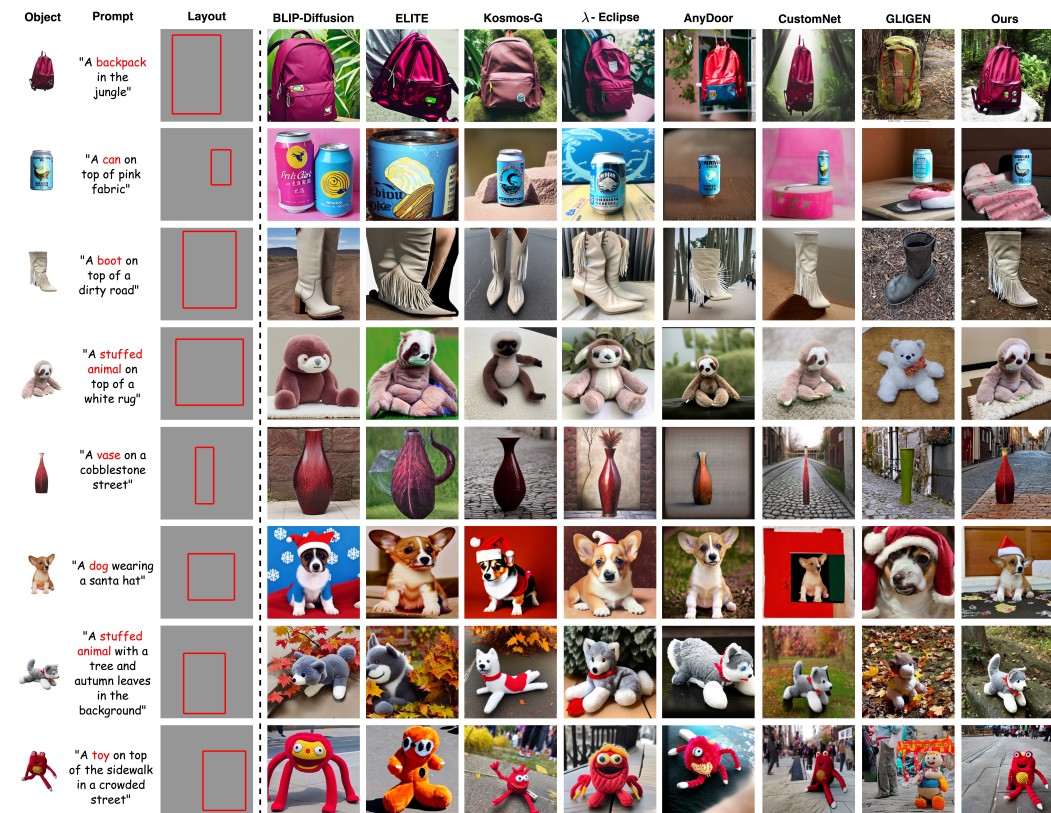

Figure 5: Visual comparison with existing methods on DreamBench objects for the single-subject customization task. Please zoom in to see the details.

from 219,188 videos across 238 object categories, with fine-grained annotations of object masks. In the data processing stage of MVImgNet, following AnyDoor (Chen et al., 2023b), for each object, we randomly selected two different frames from the same video clip to form a training pair. We apply the object mask on one frame to obtain the background-free object as the input reference object. For the other frame, we use the bounding box of the object as the grounding information and use this frame as the training ground truth. For single-image data, we use LVIS (Gupta et al., 2019), a well-known dataset for fine-grained large vocabulary instance segmentation, including 118,287 images from 1,203 categories. For each sample, we select only the object instances with top-10 largest bounding box area as training data.

**Evaluation Metrics**    We calculate the CLIP-I (Radford et al., 2021) score and DINO (Caron et al., 2021) score to evaluate the identity preservation performance of the subjects and use CLIP-T (Radford et al., 2021) score to evaluate the text alignment performance of the generated image. For evaluation of the model's grounding ability, we use $AP_{50}$ based on a pretrained YOLOv8 (Jocher et al., 2023) object detection model.

## 4.1 SINGLE SUBJECT CUSTOMIZATION

We compare our work with existing state-of-the-art works on DreamBench (Ruiz et al., 2023) for the customization of a single subject. In this experiment, we use the bounding box of the subject in the ground-truth image as the input layout. The qualitative and quantitative results are shown in Fig. 5 and Table 1, respectively. Overall, our method shows significantly better performance in layout alignment, reference object identity preservation, and background text alignment. Existing encoder-based subject-driven text-to-image customization methods BLIP-Diffusion (Li et al., 2024), ELITE (Wei et al., 2023), $\lambda$-eclipse (Patel et al., 2024) and MLLM-based method Kosmos-G (Pan et al., 2023) fail to maintain accurate identity of the reference objects lack the ability for precise layout

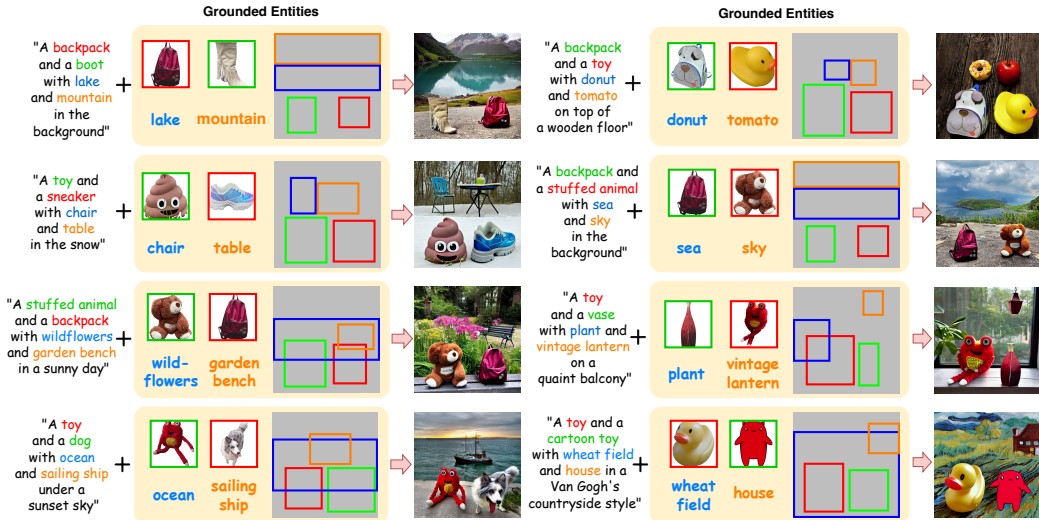

Figure 6: Multi-concept customization on DreamBench objects. Please zoom in to see the details.

control. AnyDoor (Chen et al., 2023b) is designed for image composition on a given background and lacks the ability of text-to-image generation. Although previous grounded text-to-image generation methods like GLIGEN (Li et al., 2023) are able to achieve layout control, it cannot preserve the identity of the subjects. CustomNet (Yuan et al., 2023) achieves flexible pose control. However, it highly relies on the pretrained model Zero123 (Liu et al., 2023a), limiting the resolution of its generated image to be $256 \times 256$. Moreover, there can be obvious artifacts around the boundary of the generated subject.

As an interesting observation, we find that previous non-grounding based customization methods are inclined to generate objects that are very large and in the center of the image, which gains benefit in CLIP-I score and DINO score during evaluation. However, in real-world scenarios, users may expect to flexibly control the size of the subject in the generated images. They may choose to generate larger background with broader textual information, where, in such cases, non-grounding customization meth-

Table 1: Comparison with existing methods on Dreambench.

| | CLIP-T ↑ | CLIP-I ↑ | DINO-I ↑ |
|---|---|---|---|
| SD V1.4 [(Rombach et al., 2022)] | 0.3122 | 0.8413 | 0.6587 |
| BLIP-Diffusion [(Li et al., 2024)] | 0.2824 | 0.8894 | 0.7625 |
| ELITE [(Wei et al., 2023)] | 0.2461 | 0.8936 | 0.7557 |
| Kosmos-G [(Pan et al., 2023)] | 0.2864 | 0.8452 | 0.6933 |
| lambda-eclipse [(Patel et al., 2024)] | 0.2767 | 0.8973 | 0.7934 |
| AnyDoor [(Chen et al., 2023b)] | 0.2430 | 0.9062 | 0.7928 |
| GLIGEN [(Li et al., 2023)] | 0.2898 | 0.8520 | 0.6890 |
| CustomNet [(Yuan et al., 2023)] | 0.2815 | 0.9090 | 0.7526 |
| **Ours** | 0.2881 | 0.9146 | 0.7884 |

ods cannot generate the desired result. The visual results in Fig. 5 demonstrate that our results achieves better identity preservation performance with accurate layout-alignment. We encourage the readers to view more visualizations in the Appendix.

## 4.2 MULTI-SUBJECT CUSTOMIZATION AND MULTI-ENTITY BACKGROUND GENERATION

With our proposed masked cross-attention module, our model seamlessly supports the customization of multiple subjects. Fig. 6 shows the qualitative results of the task where we customize multiple subjects and generate the image by grounding multiple text entities in the background. It can be observed that when inputting multiple subjects such as a bag and a boot, along with the layout of the background text entities such as the mountain and the lake, our model successfully generates the subjects and background with an accurate layout-alignment of each visual concept. The generated subjects preserve the their identity well. The overall generated image is well text-aligned and artifact free. Moreover, in several cases, even when the bounding boxes of the foreground objects have a large overlap with the background text entities, the model can distinguish subject-driven foreground generation from text-driven background generation, effectively avoiding the context blending.

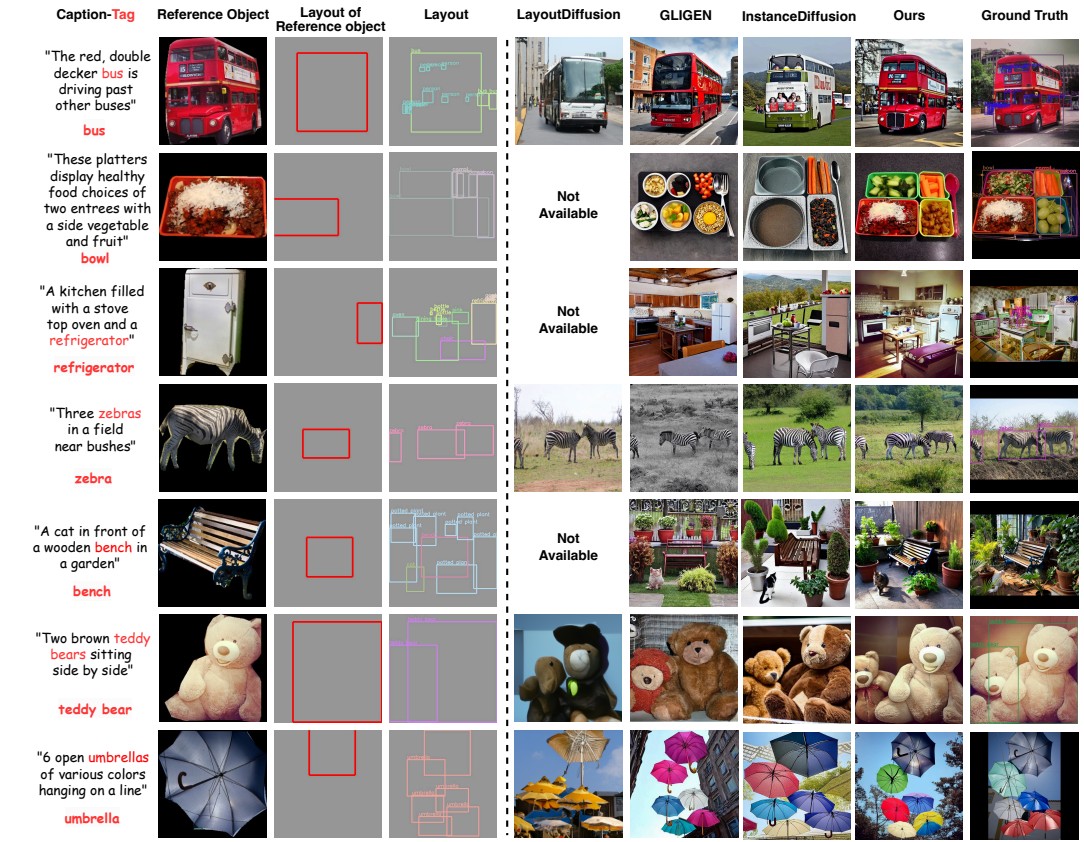

Figure 7: Visual results of reference-guided image generation with complex layout and text entities as conditions on COCO validation set. Note that LayoutDiffusion (Zheng et al., 2023) is only conducted on COCO dataset with filtered annotations, so some of its results are not available.

## 4.3 CUSTOMIZATION WITH COMPLEX LAYOUT AND TEXT ENTITIES

We evaluate our model's performance the COCO validation set for the task of generating customized images with complex layout and text entities as guidance. Quantitative and qualitative results are shown in Table 2 and Fig. 7, respectively. For each testing image, we use the largest object as the reference object (i.e., the subject), and the remaining text entities as background entities. To quantify the model's grounding ability, we adopt YOLOv8 (Jocher et al., 2023) as the object detection method. Results show that even if we input complex layouts and text entities to the model, our model can still generate high-quality scenes with precise layout alignment of all the objects and regions, and accurate identity preservation of the reference object, while preserving the text-alignment. Compared with previous layout-to-image generation methods, our model has a competitive accuracy in grounding the visual concepts and remarkable improvement on identity preservation.

As in the training stage of our model, we set the length of the max number of text tokens and the max number of image tokens to be 10 respectively, so currently the maximum number of reference subjects are set to be 10. Increasing the number of reference image tokens and text tokens will improve the maximum number of objects that the model supports, but will also increase the computation resource memory consumption and slow down the training process.

## 4.4 ABLATION STUDY

We conduct the ablation study to validate the effectiveness of our proposed components: the masked cross-attention module and the grounding module. Table 3 and Table 4 present the quantitative results on DreamBench and COCO, respectively. We observe that both the grounding module and the masked cross-attention module play a vital role in the model's grounding ability and prevent the

Table 2: Quantitative results on MS-COCO validation set for the task of customized image generation with complex layout as guidance. In this setting, we finetune our model on COCO training set, and compare with previous methods that only train on COCO.

| | CLIP-T ↑ | CLIP-I ↑ | DINO-I ↑ | $AP_{50}$ ↑ |
|---|---|---|---|---|
| LAMA[ (Li et al., 2021)] | 0.2507 | 0.8441 | 0.7330 | 18.20 |
| LayoutDiffusion[ (Zheng et al., 2023)] | 0.2738 | 0.8655 | 0.8033 | 27.40 |
| UniControl[ (Qin et al., 2023)] | 0.3143 | 0.8425 | 0.7598 | 4.53 |
| GLIGEN[ (Li et al., 2023)] | 0.2899 | 0.8688 | 0.7792 | 27.50 |
| **Ours** | 0.2946 | 0.9078 | 0.8560 | 31.10 |

Table 3: Ablation Study for modules on Dreambench.

| | CLIP-T ↑ | CLIP-I ↑ | DINO-I ↑ |
|---|---|---|---|
| *w/o* Grounding Module | 0.2762 | 0.8578 | 0.7049 |
| *w/o* Masked Cross-Attention | 0.2878 | 0.8616 | 0.7065 |
| **Full** | **0.2881** | **0.9146** | **0.7884** |

Table 4: Ablation Study for modules on MS-COCO Validation Set.

| | CLIP-T ↑ | CLIP-I ↑ | DINO-I ↑ | $AP_{50}$ ↑ |
|---|---|---|---|---|
| *w/o* Grounding Module | 0.2796 | 0.8605 | 0.7740 | 22.00 |
| *w/o* Masked Cross-Attention | 0.2884 | 0.8707 | 0.7970 | 28.50 |
| **Full** | **0.2946** | **0.9078** | **0.8560** | **31.10** |

information leakage of the reference object. Benefiting from these two modules, the model shows stronger ability of identity preservation, text alignment and grounded generation.

## 4.5 USER STUDY

In Table 5, we show the user study results comparing our model with existing models (Chen et al., 2023b; Yuan et al., 2023; Li et al., 2023) on DreamBench. Specifically, given the same input, we generate results with each model. Then we ask the users to make side-by-side comparison of our result and a randomly chosen result from the baselines regarding identity preservation, text alignment, grounding ability, and overall image quality. We collect the user responses using Amazon Mechanical Turk. Results show that participants have significantly higher preference on our method. We show details about user study in the Appendix Sec. D.

Table 5: User Study based on DreamBench: In the questions, the user is presented side-by-side comparisons of our generated image and another image randomly chosen from one of the baselines. The results in the table show user preference percentage.

| | Ours | CustomNet | Ours | AnyDoor | Ours | GLIGEN |
|---|---|---|---|---|---|---|
| Identity | **60.78** | 39.22 | **59.31** | 40.69 | **72.81** | 27.19 |
| Grounding | **56.86** | 43.14 | **64.21** | 35.79 | **58.25** | 41.75 |
| Text Alignment | **51.96** | 48.03 | **73.52** | 26.47 | **55.34** | 44.66 |
| Overall Quality | **54.41** | 45.58 | **62.25** | 37.74 | **58.74** | 41.26 |

## 5 CONCLUSION AND FUTURE WORK

We presented GroundingBooth, a general framework for the grounded text-to-image customization task. Our model has achieved a joint grounding for both reference images and prompts with precise object location and size control while preserving the identity and text-image alignment. Our strong results suggest that the proposed text-image feature grounding module and the masked cross-attention module are effective in reducing the context blending between foreground and background. We hope our research can motivate the exploration of a more identity-preserving and controllable foundation generative model, enabling more advanced visual editing.

Although our model successfully generates customized images with layout control, there are still several limitations. First, the model's performance can be limited by the base model. We can address this by using a stronger base model. Second, the design of reusing the masked cross-attention layer for each subject could still be time-consuming during inference. This can be addressed by developing a parallel generation structure for multiple subjects. We leave this direction in future work.

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

APPENDIX

## A PRELIMINARY

Our model is based on Stable Diffusion v1.4 Rombach et al. (2022), a Latent Diffusion model (LDM) that applies the diffusion process in a latent space. Specifically, an input image $x$ is encoded into the latent space using a pretrained autoencoder $z = \mathcal{E}(x), \hat{x} = \mathcal{D}(z)$ (with an encoder $\mathcal{E}$ and a decoder $\mathcal{D}$). Then the denoising process is achieved by training a denoiser $\epsilon_\theta(z_t, t, f_c)$ that predicts the added noise following:

$$\min_\theta E_{z_0, \epsilon \sim \mathcal{N}(0,1), t \sim U(1,T)} \|\epsilon - \varepsilon_\theta(z_t, t, f_c)\|_2^2, \tag{8}$$

where $f_c$ is the embedding of the condition (such as a prompt) and $z_t$ is the latent noise at timestamp $t$.

## B TRAINING/INFERENCE DETAILS

Our model is trained on 4 NVIDIA A100 GPUs for 100k steps with a batch size of 14 and a learning rate of $5 \times 10^{-5}$. During training, we randomly drop reference image embedding and text embedding both at the rate of 10%. We decently rank the area of the boxes per images, and set the max number of grounding boxes to be 10 with the largest areas. During inference, we set classifier-free guidance(CFG) (Ho & Salimans, 2022) as 3.

## C DETAILS ABOUT DATA COLLECTION

For each reference image, we use the segmentation mask to mask out the background and get the background-free reference object. In inference stage, we use SAM (Kirillov et al., 2023) to get the mask of the reference object, and get the background-free reference object.

## D DETAILS ABOUT USER STUDY

Our user study is based on DreamBench, with full 30 objects and 25 prompts. We randomly generated layouts, and use them in the generation. In the user study, given the layout, the reference object, the text prompt, the result of our method and a random-selected baseline method, we request the user to answer the following four questions:

(1) Which generated image do you think that its object is more similar to the input object? Choose between Option A and B.

(2) Which generated image do you think that its object is most likely to be at the right position as the input layout? Choose between Option A and B.

(3) Which generated image do you think is most likely to match the text description? Choose between Option A and B.

(4) Which image do you think has better image quality? Choose between Option A and B.

We received more than 1200 votes from over 530 users. In the experiment, we randomly shuffle the order of baselines to improve the confidence of the user study.

## E ADDITIONAL QUALITATIVE RESULTS ON POSE CHANGE

In Fig. 8 we show results about changing the shape of the bounding box. For grounded text-to-image customization, different from traditional text-to-image customization, the pose of the object is jointly influenced by the shape of the bounding box and the model's ability to adapt the object to be harmonious with the background. The model tend to first adapt the object to the bounding box, then make pose adjustments to make object to be harmonious with the background. For instance, in the 1st and 4th row of Fig. 8, given a bounding box with a large or small width/height ratio, the grounded customized generation will generate objects with large pose change to adapt to the bounding box,

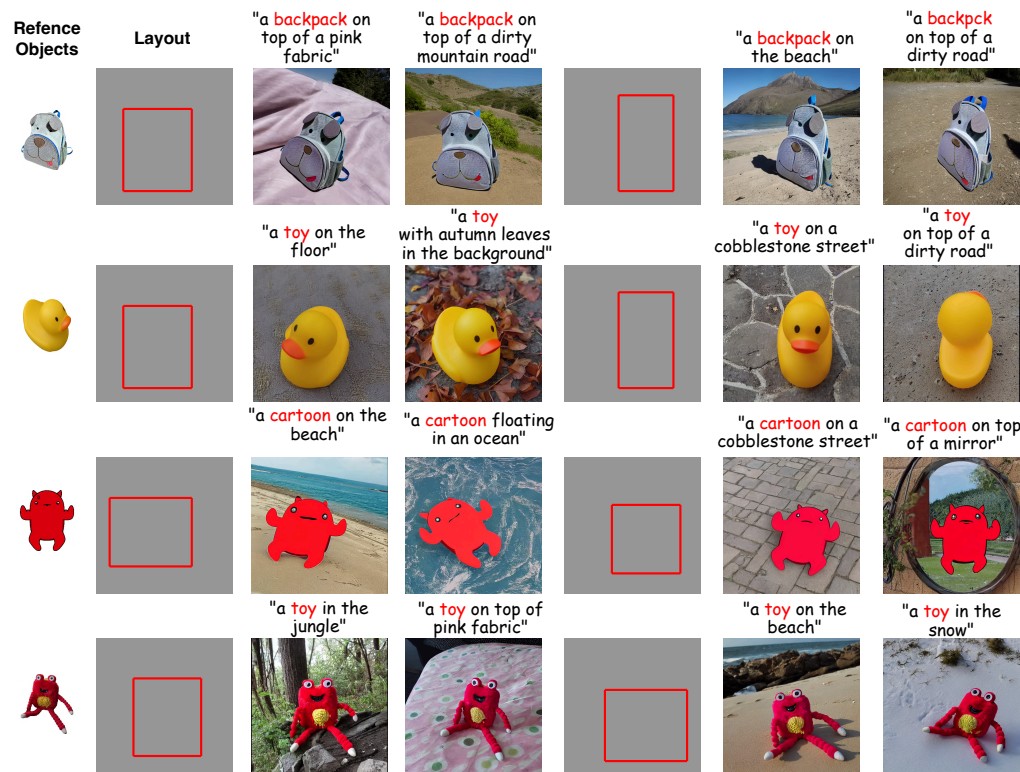

Figure 8: More visual results of our model about layout and pose change: in our model, the pose of the object is influenced by both the shape of the bounding box and the model's ability to adapt to the background. The model tends to first adapt the object into the layout, then adapt the pose to maintain harmonization with the background.

then make pose refinement inside the bounding box. Users can easily conduct the initial manipulation of the object by specifying the desired layout, then the model will automatically adjust the pose of the object to be harmonious with the background. Our model shows both the ability to generate objects with accurate location and the ability to make pose changes to the objects.

## F  ANALYSIS ON GROUNDING CIRCUMSTANCE

We also show qualitative results under the consumption that no layout is provided by the users. From the results, we can see that: Our model also supports text-to-image generation, layout-to-image generation, and personalized text-to-image generation tasks.

- As shown in Fig. 9, if the bounding box is set to be $[x1, y1, x2, y2] = [0, 0, 0, 0]$, the model will degrade into simpler text-to-image generation task, since the corresponding grounding tokens are set to be all-zero, and the model also loses the grounding ability.

- As shown in Fig. 10, if no reference object as input, and all the layouts rely on the input text entity to generate, then the model will degrade into layout-guided text-to-image generation task.

- If randomly assigned the bounding box of the reference object, our model is equal to the text-to-image personalization task, like previous non-grounding text-to-image customization works.

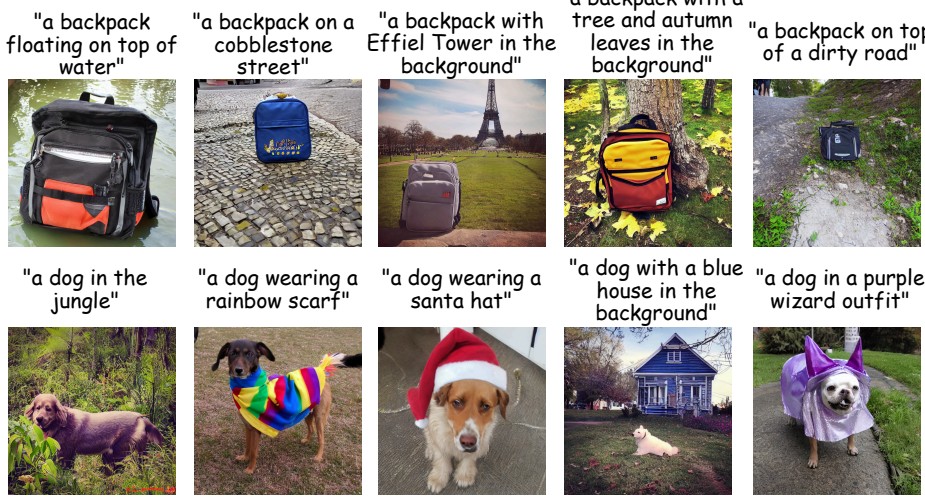

**Layout Input [0,0,0,0]**

Figure 9: Our model can also deal with pure text-to-image generation task. When we set the layout $[x1, y1, x2, y2] = [0.0, 0.0, 0.0, 0.0]$, the model will degrade into a simpler text-to-image generation task, since the corresponding grounding tokens are set to be all-zero, and the model also loses the grounding ability.

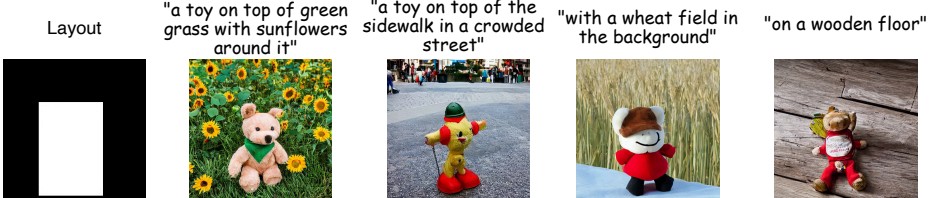

Figure 10: Our model can also deal with layout-guided text-to-image generation task: when there is no reference image input, the model will degrade into a layout-guided text-to-image generation task.

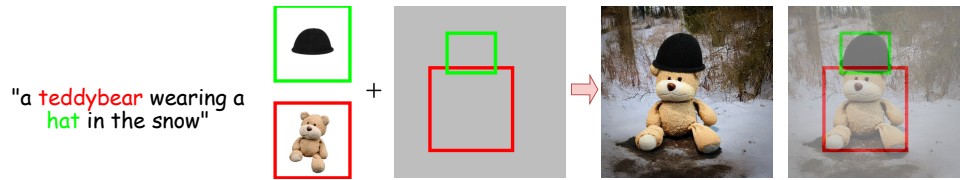

Figure 11: More results about live animals wearing clothes.

## G MORE RESULTS ABOUT OBJECT INTERACTION

As shown in Fig. 11, taking a toy object and a hat as input, our model is able to put the hat on the teddy bear, which shows the model's ability to composite reference objects.

## H MORE RESULTS ABOUT POSE CHANGE UNDER THE GUIDANCE OF PROMPT

We further show comparison results about pose change under the guidance of prompts in Fig. 12. We select prompts that is relevant to actions and pose change. Previous text-to-image customization models cannot maintain the identity of the reference object(row 2, row 4 and row 5), fail to achieve the prompt action-guided pose change(row 3 and row 4) and maintain text-alignment in certain

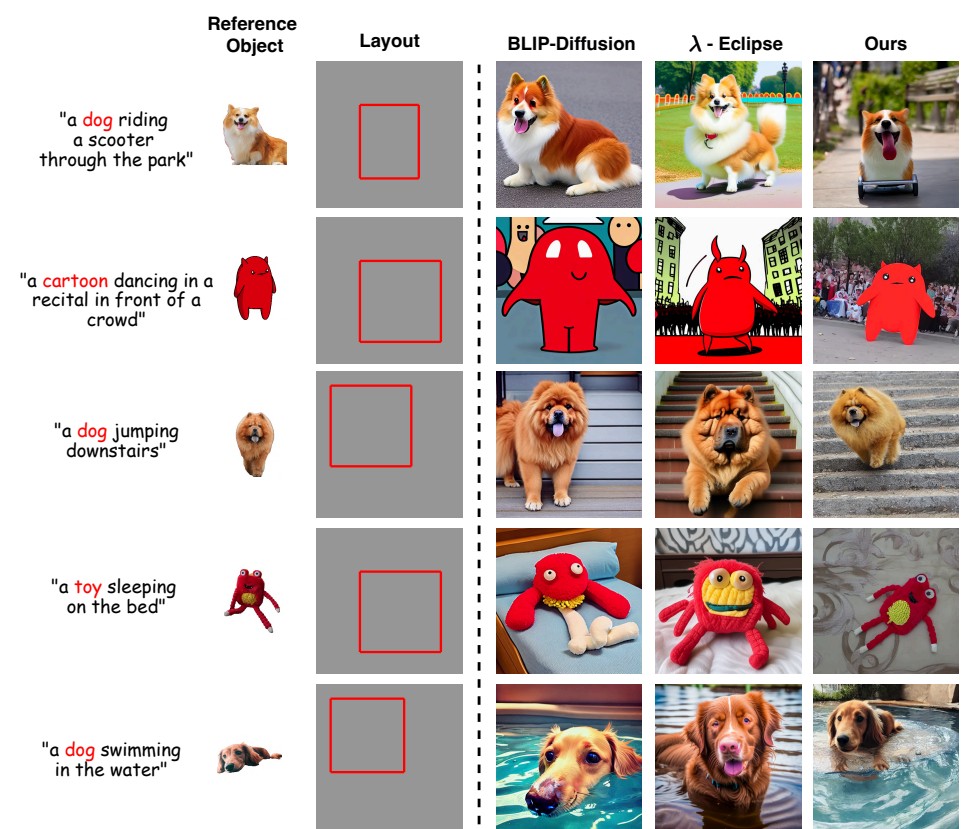

Figure 12: More results about pose change under the guidance of prompt.

Table 6: Comparison with existing methods on Dreambench under layout scale normalization.

| | CLIP-T ↑ | CLIP-I ↑ | DINO-I ↑ |
|---|---|---|---|
| SD V1.4 [(Rombach et al., 2022)] | 0.3122 | 0.8413 | 0.6587 |
| BLIP-Diffusion [(Li et al., 2024)] | 0.2824 | 0.8894 | 0.7625 |
| ELITE [(Wei et al., 2023)] | 0.2461 | 0.8936 | 0.7557 |
| Kosmos-G [(Pan et al., 2023)] | 0.2864 | 0.8452 | 0.6933 |
| lambda-eclipse [(Patel et al., 2024)] | 0.2767 | 0.8973 | 0.7934 |
| AnyDoor [(Chen et al., 2023b)] | 0.2430 | 0.9062 | 0.7928 |
| GLIGEN [(Li et al., 2023)] | 0.2898 | 0.8520 | 0.6890 |
| CustomNet [(Yuan et al., 2023)] | 0.2821 | 0.9103 | 0.7587 |
| **Ours** | **0.2911** | **0.9169** | **0.7950** |

cases(row 1 and row 3). Our method not only achieve grounded text-to-image customization, but also able to maintain a good balance between identity preservation and text alignment, and can also generate objects with variations in pose.

## I COMPARISON UNDER LAYOUT SCALE NORMALIZATION

We further conducted experiments to normalize our bounding box scales based on the average size of objects generated by other personalized text-to-image generation methods. We update the comparison results in the Table 6. For non-grounding-based text-to-image customization methods, we used Grounding DINO (Liu et al., 2023b) to detect the bounding box of the target subject by identifying the object name. We then computed the average bounding box area and applied a $\pm 20\%$ variation as the normalized bounding box size. This normalized bounding box size scale was subsequently employed for the grounded text-to-image customization methods(CustomNet (Yuan et al., 2023) and

Ours). The results demonstrate that our method achieves improved CLIP-T, CLIP-I and DINO-I scores, outperforming all baseline personalized text-to-image generation methods and layout-guided text-to-image generation methods in this case.

## J    ADDITIONAL QUALITATIVE RESULTS

Here we show more qualitative results. In Fig. 13 we show results on DreamBench and in Fig. 14 and Fig. 15 we show more results about complex background background evaluation on coco validation set.

## K    SOCIAL IMPACT

GroundingBooth provides a flexible method for users to precisely customize the layout of both foreground and background objects based on user-provided reference subjects and text descriptions without any test-time finetuning. The support for the generation of multi-subjects provides a useful tool for users to generate images using their desired layout. Users can optionally choose reference objects or simple text inputs to generate their desired image, which significantly expands the flexibility in controllable and customized text-to-image generation.

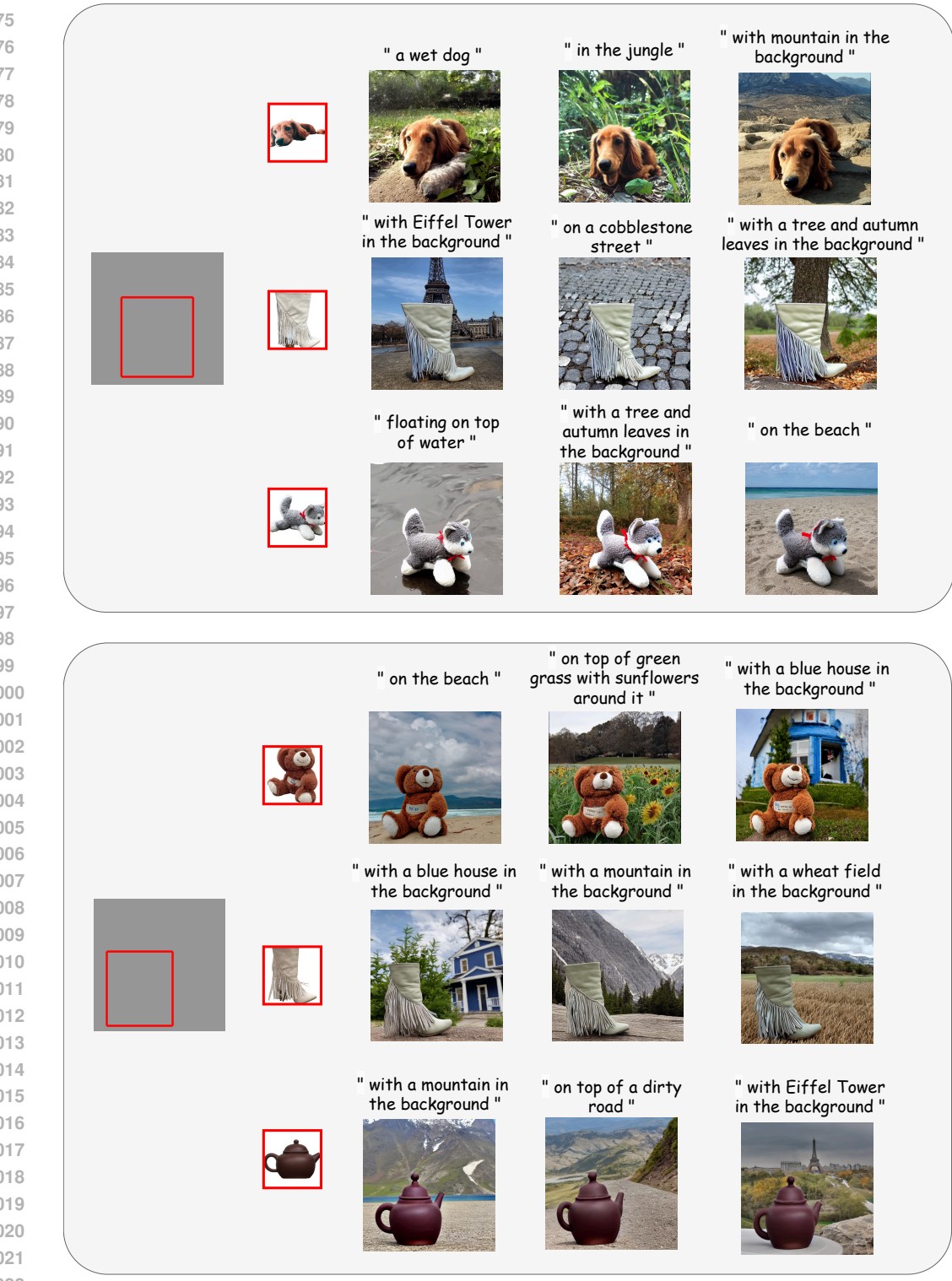

Figure 13: More visual results of our model.

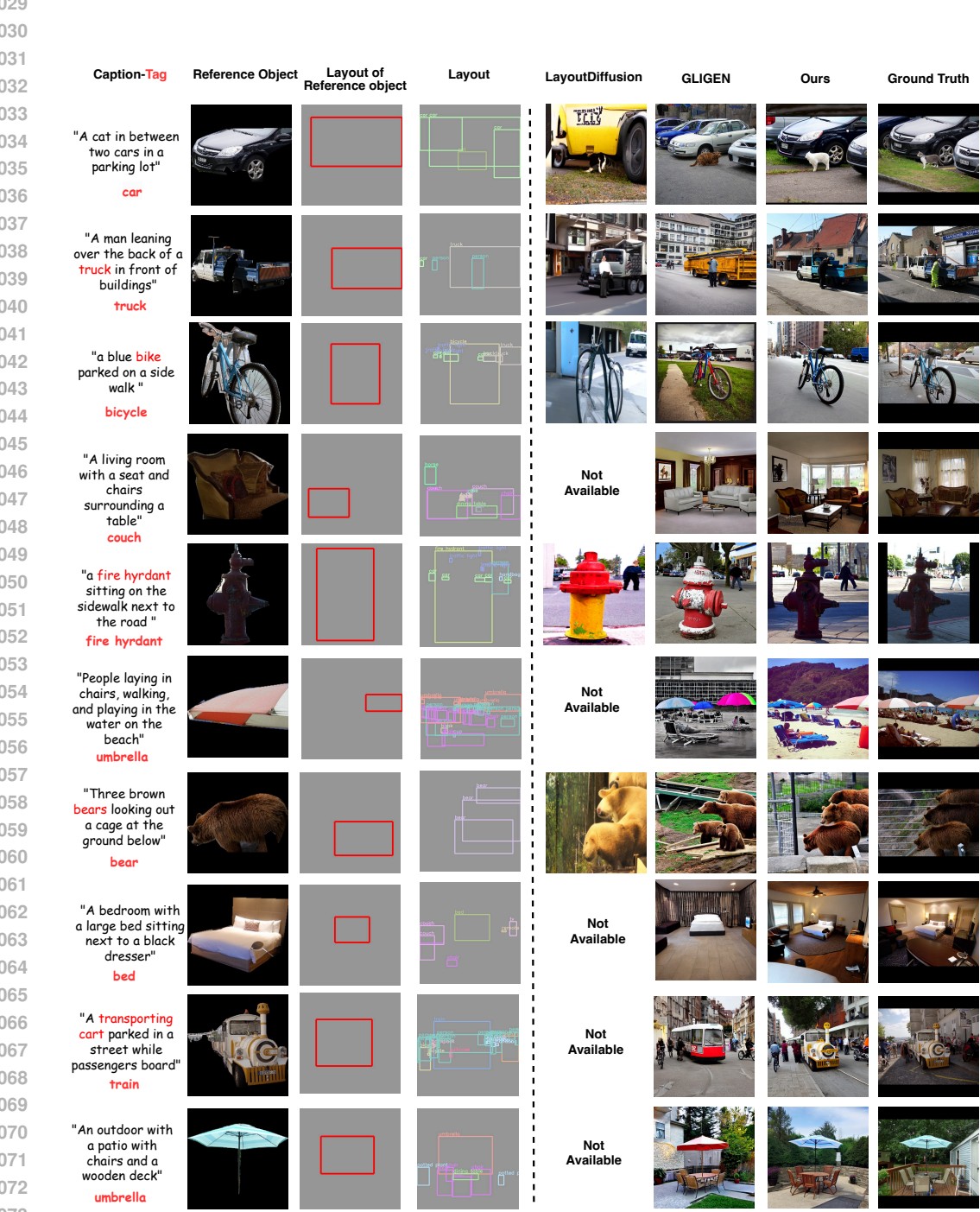

Figure 14: More results on complex scene generation on COCO validation set.

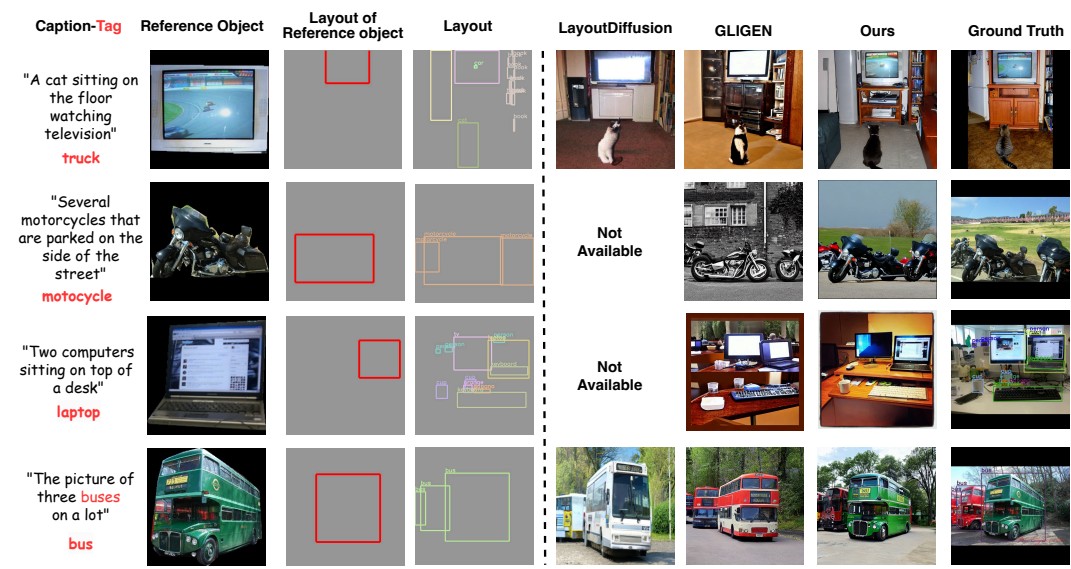

Figure 15: More results on complex scene generation on COCO validation set.

