# OpenReview forum: "GroundingBooth: Grounding Text-to-Image Customization"
_ICLR.cc/2025/Conference — Submitted to ICLR 2025_

### Official Review · Reviewer_hmpM · 2024-10-31

**Soundness:** 2
**Presentation:** 3
**Contribution:** 2
**Rating:** 6
**Confidence:** 3

**Summary:**

This paper focuses on improving the accurate generation of spatial relationships between objects and backgrounds when creating personalized object variants. Technically, the authors propose a joint text-image grounding module that encourages both foreground subjects and background objects to adhere to locations defined by input bounding boxes. They also introduce a masked cross-attention layer aimed at preventing the unintended blending of multiple visual concepts in the same location, producing clear, distinct objects. Experiments are conducted on the MVImgNet and LVIS datasets.

**Strengths:**

1. The paper tackles the task of generating personalized objects based on specific locations, which is an interesting setup.
2. This work proposes integrating reference objects and their location prompts through a grounding module and masked cross-attention.
3. Experiments are conducted on two benchmarks, accompanied by illustrative visualizations.

**Weaknesses:**

1. The paper primarily focuses on enabling the location-controlled generation of personalized objects, a setting already explored in prior work [3], which the authors seem to overlook. Additionally, the authors introduce a rather complex module to integrate location information but seem to lose focus on core functionalities like layout-to-image generation or personalized object generation.
2. Missing References: Some relevant references in layout-to-image generation, such as [1,2] and subject-driven image generation [4], are absent.
3. There are some limitations in model design. For example, the authors note that in cases where bounding boxes belong to the same class, the model cannot distinguish between a bounding box for a reference object and one for a text entity, leading to misplacement of the reference object. However, the paper does not clarify whether or how the proposed masked cross-attention module addresses this issue.
4. Further analysis is needed on topics such as the maximum number of reference objects supported in a single input and the model’s performance on subject-driven image generation without layout information.


[1] LayoutGPT: Compositional Visual Planning and Generation with Large Language Models
[2] Layoutllm-t2i: Eliciting layout guidance from llm for text-to-image generation
[3] Training-Free Layout Control With Cross-Attention Guidance
[4] Blip-diffusion: Pre-trained subject representation for controllable text-to-image generation and editing

**Questions:**

1. Does this work support simpler text-to-image generation, layout-to-image, or personalization tasks?

2. Regarding the illustration of the masked cross-attention layer in Figure 2, is the number of layers determined by the number of reference objects? For example, if there are three reference objects in the input, does that mean three masked cross-attention modules are required? If so, this model design seems unreasonable. Sequential masking could result in information loss in subsequent modules, especially when reference objects have significant overlap.

---

> ### Author Response · Authors · 2024-11-21
> **Response to Reviewer hmpM**
>
> Dear Reviewer QbKw:
>
> Thanks a lot for your insightful feedback and kind advice. We have make some updates in the revised PDF. We would like to address your concerns one by one as follows:
>
> 1. About comparison with prior works:
>
> Thanks for pointing it out. We have cited this paper in the revised version of the paper. However, we need to emphasize that our task is subject-driven customized text-to-image generation, while [3] is simply text-to-image generation,  it does not allow reference objects as input, and cannot maintain the identity of the reference objects. These are different tasks. Our method achieves personalized generation with layout guidance.
>
> 2. About missing references:
>
> Thanks for your suggestion. I have cited these papers in the revised version of the paper. For [4] Blip-Diffusion, we have not only cited but also compared with their method in **Table 1** and **Fig.5**.
>
> 3. How does the proposed masked cross-attention module distinguish between a bounding box for a reference object and one for a text entity:
>
> We use an example to explain this in **Fig.4** of the revised version of the paper. In **Fig.4**, both the reference objects and the text entities have cats and dogs. The model can distinguish whether each bounding box belongs to the same class and effectively avoids the misplacement of the generated objects. The masked cross-attention module allows using the bounding box to restrict the injection of the reference object information inside the target bounding box.  The masked cross-attention is conducted on the DINO Image feature space, which helps distinguish whether a bounding box is for a reference object or for a text entity.
>
> 4. Further analysis is needed on topics such as the maximum number of reference objects supported in a single input and the model’s performance on subject-driven image generation without layout information:
>
> As in the training stage of our model, we we set the length of the max number of text tokens and the max number of image tokens to be 10, so currently the maximum number of reference subjects is set to be 10. Increasing the number of reference image tokens and text tokens will improve the maximum number of objects that the model supports, but will also increase the computation memory consumption and slow down the training process. We add this to **L475-L479** of the revised version of the paper.
>
> For the circumstance of no layout-guidance, please see the explanation below in A1:
>
> Q1: Does this work support simpler text-to-image generation, layout-to-image, or personalization tasks?
>
> A1: Definitely yes.  We show further experiments of this in  **Fig.9** and **Fig.10** of the Appendix.
>
> (1) If the bounding box is set to be [x1,y1,x2,y2] = [0,0,0,0], the model will degrade into a simpler text-to-image generation task, since the corresponding grounding tokens are set to be all-zero, and the model also lose the grounding ability. Please see **Fig. 9.** in the Appendix.
>
> (2) If no reference object as input, and all the layouts rely on the input text entity to generate, then the model will degrade into a pure layout-guided text-to-image generation task. See **Fig.10** in the Appendix.
>
> (3) If randomly assigned the bounding box of the reference object, our model is equal to the text-to-image customization task, like previous non-grounding text-to-image customization works.
>
> Q2: About model design:
>
> A2: Thanks for your suggestion. We have mentioned this in the future work of the submission **L537-L539**. We will address these in our future works.
>
> During the training stage, we only train a single masked cross-attention layer. During inference, this masked cross-attention layer is reused for each subject. In **Fig.2**, we would like to express that for each subject, during inference, we reuse the same layer but **not** introduce new layers. This can prevent the semantic mis-blending of the visual contents, especially in the overlapping regions.
>
> Reference:
>
> [1] Feng W, Zhu W, Fu T, et al. LayoutGPT: Compositional Visual Planning and Generation with Large Language Models
>
> [2] Qu L, Wu S, Fei H, et al. Layoutllm-t2i: Eliciting layout guidance from llm for text-to-image generation
>
> [3] Chen M, Laina I, Vedaldi A. Training-Free Layout Control With Cross-Attention Guidance
>
> [4] Li D, Li J, Hoi S. Blip-diffusion: Pre-trained subject representation for controllable text-to-image generation and editing

---

> > ### Author Response · Authors · 2024-11-23
> >
> > Dear Reviewer,
> >
> > We hope our responses have adequately addressed your previous concerns about (1) comparison with prior works, (2) details about masked cross-attention, (3) analysis about the max number of objects the model supports, (4) the model's generalization ability to other tasks and (5) details of model design. We really look forward to hearing from you and would be happy to discuss and address any remaining concerns that you may still have.
> >
> > Thanks,
> >
> > Authors

---

> > > ### Author Response · Authors · 2024-11-24
> > > **Kindly Reminder**
> > >
> > > Dear Reviewer hmpM,
> > >
> > > We sincerely appreciate the time and effort you have dedicated to reviewing our submission. We have submitted our response and would like to follow up to inquire whether our response has sufficiently addressed your concerns.
> > >
> > > Please do not hesitate to let us know if you have any remaining questions or require additional clarification. We look forward to hearing from you. We are eager to address them promptly before the discussion deadline.
> > >
> > > Thank you once again for your valuable insights and guidance.
> > >
> > > Best regards,
> > >
> > > GroundingBooth Authors

---

> > > > ### Comment · Reviewer_hmpM · 2024-11-25
> > > >
> > > > Thanks for the response. I still have the following questions:
> > > >
> > > > 1. To my understanding, [3] also supports subject-driven text-to-image generation, as a real image can be taken as input in Figures 1 and 8.
> > > > 2. The model design part is still not clear. During inference, is reuse performed sequentially or in parallel?
> > > >
> > > > Moreover, after carefully reviewing the feedback from other reviewers, I noticed that there are still many aspects of the work that require revisions. Additionally, it remains unclear whether these issues have been fully addressed.
> > > >
> > > > Therefore, I will maintain my current score.

---

> ### Author Response · Authors · 2024-11-27
> **Response to Reviewer hmpM**
>
> Thanks for your response. We will answer one by one.
>
> 1. About differences between our method and [3]:
>
> (1) [3] employs a test-time fine-tuning approach, specifically relying on the fine-tuning of Dreambooth[4] at inference. Test-time fine-tuning methods finetune a pretrained diffusion model on a few subject images so that the model is adapted to a new identifier token representing the new concept. In contrast, our method is encoder-based, which makes it significantly more efficient during inference, providing much faster image customization. Also, our encoder-based methods are usually more robust to hyper-parameters compared with training-free methods, and can be easily generalized to unseen objects and scenarios.
>
> (2) Our implementation approach also differs significantly. While [3] utilizes a backward guidance mechanism, our method incorporates a joint grounding module and masked cross-attention to enable layout-guided generation, allowing for a more structured and effective integration of guidance signals.
>
> (3)  [3] is limited to use layout to control the generation of foreground object. In contrast, our method is capable of achieving joint subject-driven foreground and text-driven background layout control, allowing for more comprehensive customized scene manipulation.
>
> 2. About layer reuse:
>
> During inference, the reuse of masked cross-attention is conducted in a sequential manner. Currently, there is no conclusive evidence demonstrating the superiority of either sequential or parallel cross-attention mechanisms in this context. Sequential attention models have been effectively employed in interactive image generation and image editing tasks [1][2], demonstrating their efficacy and robustness. Moreover, the sequential approach is well-suited for extending to grounded text-to-image customization tasks, facilitating multi-subject grounded customization with greater flexibility.
>
> In our experiments on multi-subject grounded text-to-image customization, we observed that injecting the DINO features of all reference objects into the same layer leads to context blending in overlapping regions between bounding boxes. By contrast, sequentially injecting the features makes it easier to delineate visual concepts and prevents context blending, as the newly injected feature replaces the previous one in overlapping regions. This sequential injection approach demonstrated superior performance for multi-subject grounded text-to-image customization, effectively preserving the distinct characteristics of each subject.
>
> Please feel free to reach out if you have any further questions or need additional clarification.
>
>
> [1] Cheng Y, Gan Z, Li Y, et al. Sequential attention GAN for interactive image editing, ACM MM 2020
>
> [2] Guo Q, Lin T. Focus on your instruction: Fine-grained and multi-instruction image editing by attention modulation, CVPR 2024
>
> [3] Chen M, Laina I, Vedaldi A. Training-Free Layout Control With Cross-Attention Guidance. WACV 2024
>
> [4] Ruiz N, Li Y, Jampani V, et al. Dreambooth: Fine tuning text-to-image diffusion models for subject-driven generation. CVPR 2023

---

> > ### Author Response · Authors · 2024-12-01
> > **Kindly Reminder**
> >
> > Dear Reviewer hmpM,
> >
> > As the deadline draws near, we would like to kindly follow up to inquire whether our response has sufficiently addressed your concerns. We have submitted our response, and we are still eager to engage in further discussion if you have any additional concerns or suggestions.
> >
> > Additionally, we would like to mention that other reviewers have provided updated feedback and made adjustments to their scores. If our response has sufficiently addressed your concerns, we would be truly appreciative if you could consider reflecting that in your evaluation as well.
> >
> > Thank you for your time and thoughtful consideration.
> >
> > Best regards,
> >
> > GroundingBooth Authors

---

> ### Comment · Reviewer_hmpM · 2024-12-02
>
> Thanks for the authors' response. Most of my main concerns have been addressed. After reviewing the overall content of the paper, I will raise my score to 6.
>
> Additionally, I’d like to further clarify the following:
>
> 1. How should we understand the claim: "encoder-based methods are usually more robust to hyper-parameters compared with training-free methods, and can be easily generalized to unseen objects and scenarios"? Could you please provide a more detailed explanation?
>
> 2. Where exactly in the paper can I find the experiments on multi-subject grounded text-to-image customization?
>
> Thanks!

---

> > ### Author Response · Authors · 2024-12-03
> >
> > Dear Reviewer hmpM:
> >
> > Thanks for your response!
> >
> > 1. Further explanation about robustness and generalization ability:
> >
> > Training-free customized image editing methods such as PhotoSwap [1] and SwapAnything [2] introduce several additional hyper-parameters in the inference stage that need careful adjustment. In practice, we found these methods to be highly sensitive to these hyper-parameters, requiring dedicated tuning for each individual test case to achieve the desired result.
> >
> > On the other hand, for pretrained methods (i.e., encoder-based methods), they train a generalizable diffusion model on a large-scale dataset. These methods do not require additional hyper-parameter adjustments during inference. As the model is pretrained on a large-scale dataset covering a wide range of objects and scenarios, it can generalize effectively to unseen objects and conditions during testing, without further tuning the hyper-parameters. The encoder-based approach is evidently more flexible and can generalize easily to novel subjects without additional computational cost in tuning the hyper-parameters.
> >
> > 2. The multi-subject grounded text-to-image customization is shown in both Fig.1(b) and Fig.6 in the revised Pdf(Fig.1(b) and Fig.5 in the original version).
> >
> > Best regards,
> >
> > GroundingBooth Authors
> >
> > [1] Gu J, Wang Y, Zhao N, et al. Photoswap: Personalized subject swapping in images. NeurIPS 2023.
> >
> > [2] Gu J, Zhao N, Xiong W, et al. Swapanything: Enabling arbitrary object swapping in personalized visual editing. ECCV 2024.

---

### Official Review · Reviewer_QbKw · 2024-11-02

**Soundness:** 3
**Presentation:** 4
**Contribution:** 2
**Rating:** 3
**Confidence:** 4

**Summary:**

The paper presents GroundingBooth, a method for grounded text-to-image customization. Given a list of subject entities represented by images and text entities represented by textual descriptions, along with bounding-box locations, GroundingBooth aims to generate an image containing all subjects in the specified locations according to their bounding boxes.

**Strengths:**

* The authors tackle the important task of grounded image generation with both text and image localization conditions.
* The writing is clear, making it easy to understand the proposed method.
* The authors combine grounded generation from both reference objects and textual inputs within a single architecture, which is highly relevant for many applications.
* The authors evaluate their method against a variety of prior works and datasets.

**Weaknesses:**

* In all the qualitative examples, the generated objects remain in the same pose as in the input image, despite the claim in line 191: “Moreover, our work adaptively harmonizes the poses of the reference objects and faithfully preserves their identity.” Could you provide examples where the input subjects change their pose while maintaining their identity? I would like to see examples where the prompt requires a significant pose change from the input subject.

* The proposed Masked Cross-Attention module was presented in previous works; see, for instance:
[1] Be Yourself: Bounded Attention for Multi-Subject Text-to-Image Generation, Dahary et al. ECCV 2024
[2] InstanceDiffusion: Instance-level Control for Image Generation, Wang et al. CVPR 2024

* Overall, the proposed modules seem to lack novelty. The gated self-attention mechanism is borrowed from GLIGEN, and the masked cross-attention module exists in prior work, such as in [1].

* I find the distinction between “background” and “foreground” objects confusing, as it actually separates objects based on their source (image or text) rather than their position in the background or foreground of the image.

* The quantitative results are not convincing, as GroundingBooth shows lower scores than prior work on several metrics (e.g., Tables 1 and 2).

**Questions:**

* For personalization of a single subject (Fig. 4, Table 1), how is the bounding box determined? How do you compare with methods that do not require a bounding box as input?
* How well can the method generate interactions between input subjects? For example, could it make the teddy bear wear the red backpack?

---

> ### Author Response · Authors · 2024-11-21
> **Response to Reviewer QbKw (1/2)**
>
> Dear Reviewer QbKw:
>
> Thanks a lot for your insightful feedback and kind advice. We would like to address your concerns one by one.
>
> 1. Examples where the input subjects change their pose while maintaining their identity:
>
> We further provide examples of the pose change of the input subjects in **Fig.8** of the Appendix. For grounded text-to-image customization, the pose change is more complex than the non-grounding customization methods. From the experiments, we find that the pose is influenced by both the shape of the bounding box and the model’s ability to adapt to the background. The model tends to first adapt the object into the bbox, then adapt the pose to maintain harmonization with the background. Adjusting the shape of the bounding box will lead to a large pose change. Also within the same bounding box, the model has learned to adjust the object’s pose to be harmonious with the generated background. For instance, in **Fig.8**, given a bounding box with a large or small width/height ratio, the grounded customized generation will generate objects with large pose changes to adapt to the bounding box, then make refinement inside the bounding box. Users can easily conduct the initial manipulation of the object by specifying the desired layout, then the model will automatically customize the background. Our model shows both the ability to generate objects within the correct location and make pose changes to ensure harmonious integration with the scene.
>
> 2. About the difference of our Masked Cross-Attention compared with other methods:
>
> First, we would like to clarify that both Be-Yourself and InstanceDiffusion are text-to-image generation methods. They cannot do customized text-to-image generation tasks, which do not support the input of the reference object and fail to maintain the identity of the reference object.  Using a mask in the cross-attention of the transformer blocks has been proven effective in grounded generation while the detailed forms are different, and it is natural to adopt it in grounded text-to-image customization tasks. There are differences between our method and these methods:
>
> (1) These two methods directly apply masks on the text embedding attention maps, while our method uses a coarse-to-refine method. We first inject clip text embedding to the attention map through cross-attention to generate all the visual contents, then use masked cross-attention on the DINO image embeddings to refine the feature within the box of the subject. Through the coarse-to-fine method, the model can inject image features in the attention blocks while restricting the injection of the image embedding to refine the feature map inside the corresponding bbox.
>
> (2) Be-yourself uses a time-specific mask in both self-attention and cross-attention layers. InstanceDiffusion uses a masked self-attention and fusion method, while our method uses masked cross-attention.
>
> 3. About the “distinction between “background” and “foreground” objects”:
>
> Thanks for pointing out. The original idea of our paper is that foreground refers to the reference objects and the background refers to the text entities. Actually users can flexibly specify the box position and assign reference image/text prompts to each box. In practical usage, users tend to assign the boxes of the reference objects in the front as the foreground. We will make these statements clear in the revised version.
>
> 4. About questions for the quantitative results:
>
> We emphasize that our task is fine-grained **subject-driven text-to-image customization**. It's not merely a combination of layout-guided text-to-image generation and personalized text-to-image generation. Our method carefully balances identity preservation, text alignment, layout alignment, pose change, and harmonization. Our approach shows competitive results in quantitative evaluations and enables flexible grounded text-to-image customization.
>
> In essence, the grounded text-to-image customization task requires balancing identity preservation, layout, and text alignment. Our experiments in **Fig. 5** show that these methods tend to generate results with large-scale objects in the images. During evaluation, larger bounding boxes benefit the DINO-I score and CLIP-I score, as larger objects typically maintain more detailed features of the reference objects. We have elaborated on this in paper **Sec 4.1** **L404-L418**.
>
> Additionally, the shape and size of the reference object's bounding box influence the results. Bounding boxes with notably large or small height-to-width ratios affect the evaluation of identity preservation.

---

> ### Author Response · Authors · 2024-11-21
> **Response to Reviewer QbKw (2/2)**
>
> 5. How is the bounding box determined and how is it compared with other methods?
>
> In **Table 1**, we specify the bounding box to be the same as the bounding box of the object in the reference image; and in **Fig. 5** in the revised pdf (**Fig. 4** in the original version), we use the same set of random bounding boxes of a range of scales. For other non-grounding-based methods, we just do not take a bounding box as input.
>
> 6: How well can the method generate interactions between input subjects?
>
>  We show some further visualizations in the Appendix **Fig. 11** of the revised PDF. Results who that our model can put a hat on the teddy bear, which shows that our model can deal with the interactions between reference objects.

---

> > ### Author Response · Authors · 2024-11-23
> >
> > Dear Reviewer,
> >
> > We hope our responses have adequately addressed your previous concerns about (1) pose change of the objects, (2) masked cross-attention, (3) the distinction between "background" and "foreground", (4) quantitative results, (5) the determine of the bounding boxes in experiments and (6) interactions between input subjects. We really look forward to hearing from you and would be happy to address any remaining concerns that you may still have.
> >
> > Thanks,
> >
> > Authors

---

> > > ### Author Response · Authors · 2024-11-24
> > > **Kindly Reminder**
> > >
> > > Dear Reviewer QbKw,
> > >
> > > We sincerely appreciate the time and effort you have dedicated to reviewing our submission. We have submitted our response and would like to follow up to inquire whether our response has sufficiently addressed your concerns.
> > >
> > > Please do not hesitate to let us know if you have any remaining questions or require additional clarification. We look forward to hearing from you. We are eager to address them promptly before the discussion deadline.
> > >
> > > Thank you once again for your valuable insights and guidance.
> > >
> > > Best regards,
> > >
> > > GroundingBooth Authors

---

> > > > ### Comment · Reviewer_QbKw · 2024-11-24
> > > >
> > > > Thank you for the authors’ response. I have two follow-up questions:
> > > >
> > > > **Answer 1:** The results in Fig. 8 are still unsatisfactory, as the pose of all generated images remains almost identical to that of the input image. In Fig. 8, most of the variation arises from rigid object rotations. However, the cartoon character consistently raises its hands in the same way, and the red toy always sits in the same pose with its legs spread to the sides. Moreover, the prompts used in these figures do not address pose changes at all, focusing only on background modifications. I was expecting prompts that emphasize variations in the object's pose. Some potential examples include:
> > > >
> > > > Red toy: "A toy dancing in a recital in front of a crowd."
> > > > Fluffy dog (Fig. 1): "A dog cooking a gourmet meal in the kitchen."
> > > > Corgi: "A dog riding its bicycle through the park."
> > > >
> > > > In its current form, Fig. 8 raises doubts about the model's ability to handle pose-changing prompts effectively.
> > > >
> > > > **Answer 4:**  Since your method can control the size of the object, couldn’t you address the issue of object size calibration by defining the input bounding boxes to match the average size of objects generated by personalized text-to-image methods?

---

> ### Author Response · Authors · 2024-11-26
> **Response to Reviewer QbKw**
>
> Thanks for your response. We have modified in the revised version of submission. We would like to address your concerns one by one.
>
> 1: About prompt-guided pose change:
>
> We further show comparison results about pose change under the guidance of prompts in **Fig.12** of the appendix of the revised PDF.. We select prompts that is relevant to actions and pose change. Previous text-to-image customization models cannot maintain the identity of the reference object(row 2, row 4 and row 5), fail to achieve the prompt action-guided pose change(row1, row 3 and row 4) and maintain text-alignment in certain cases(row 1 and row 3). Our method is not only able to achieve grounded text-to-image customization, but also able to maintain a good balance between identity preservation and text alignment.
>
> 2. About evaluation based on layout size normalization:
>
> Thanks for your suggestion. We further conducted experiments to normalize our bounding box scales based on the average size of objects generated by other personalized text-to-image generation methods. The updated comparison results are presented in **Table A** below. For non-grounding-based text-to-image customization methods, we used Grounding DINO[1] to detect the bounding box of the target subject by identifying the object name. We then computed the average bounding box area and applied a ±20% variation as the normalized bounding box size. This normalized bounding box size scale was subsequently employed for the grounded text-to-image customization methods(CustomNet[2] and our approach). The results demonstrate that our method achieves improved CLIP-T, CLIP-I and DINO-I scores, outperforming all baseline personalized text-to-image generation methods and layout-guided text-to-image generation methods in this case.
>
> Table A: Comparison with existing methods on Dreambench under layout scale normalization.
>
> |                | CLIP-T ↑   | CLIP-I ↑   | DINO-I ↑   |
> |----------------|------------|------------|------------|
> | BLIP-Diffusion | 0.2824     | 0.8894     | 0.7625     |
> | ELITE          | 0.2461     | 0.8936     | 0.7557     |
> | Kosmos-G       | 0.2864     | 0.8452     | 0.6933     |
> | lambda-eclipse | 0.2767     | 0.8973     | 0.7934     |
> | AnyDoor        | 0.2430     | 0.9062     | 0.7928     |
> | GLIGEN         | 0.2898     | 0.8520     | 0.6890     |
> | CustomNet      | 0.2821     | 0.9103     | 0.7587     |
> | **Ours**       | **0.2911** | **0.9169** | **0.7950** |
>
> We are still eager to engage in further discussion and address any additional concerns you may have. We would be very grateful if you could raise your rating accordingly.
>
> [1] Liu S, Zeng Z, Ren T, et al. Grounding dino: Marrying dino with grounded pre-training for open-set object detection. ECCV 2024
>
> [2] Yuan Z, Cao M, Wang X, et al. Customnet: Zero-shot object customization with variable-viewpoints in text-to-image diffusion models. ACM MM 2024.

---

> ### Author Response · Authors · 2024-11-27
>
> Dear Reviewer QbKw,
>
> With the deadline for manuscript revisions approaching in less than a day, we would like to kindly follow up on the concerns you previously raised. We have provided detailed responses to address your feedback. If there are any remaining issues or areas that need further clarification, please do let us know. We greatly value your insights and are committed to ensuring the final manuscript aligns with your expectations.
>
> Thank you for your time and thoughtful consideration.
>
> Best regards,
>
> GroundingBooth Authors

---

> > ### Author Response · Authors · 2024-11-29
> > **Kindly Reminder**
> >
> > Dear Reviewer QbKw,
> >
> > We sincerely appreciate the time and effort you have dedicated to reviewing our submission. We have submitted our response and would like to follow up to inquire whether our response has sufficiently addressed your concerns.
> >
> > Please feel free to let us know if you have any remaining questions or if further clarification is needed. We are eager to address them promptly before the discussion deadline.
> >
> > Thank you once again for your valuable insights and guidance.
> >
> > Best regards,
> >
> > GroundingBooth Authors

---

> > > ### Author Response · Authors · 2024-12-01
> > > **Kindly Reminder**
> > >
> > > Dear Reviewer QbKw,
> > >
> > > As the deadline draws near, we would like to kindly follow up to inquire whether our response has sufficiently addressed your concerns. We have submitted our response, and we are still eager to engage in further discussion and address any additional concerns you may have. If you find our response addressed your concern, we would deeply appreciate it if you could consider raising our rating.
> > >
> > > Thank you for your time and thoughtful consideration.
> > >
> > > Best regards,
> > >
> > > GroundingBooth Authors

---

> ### Author Response · Authors · 2024-12-02
> **Final Day for Discussion: We look forward to your response**
>
> Dear Reviewer QbKw,
>
> With the deadline for reviewers to post messages approaching in less than a day, we would like to kindly follow up to see if our response has adequately addressed your concerns. We have submitted our detailed response and are still eager to engage in further discussion if you have any additional questions or suggestions.
>
> If our response has satisfactorily resolved your concerns, we would be truly appreciated if you could consider raising your rating.
>
> Thank you once again for your time and thoughtful consideration.
>
> Best regards,
>
> GroundingBooth Authors

---

> > ### Comment · Reviewer_QbKw · 2024-12-02
> >
> > Dear Authors,
> > Thank you for your additional response.
> >
> > **Quantitative Comparison:** Thank you for conducting the experiment presented in Table A. Since AnyDoor is also a grounded text-to-image customization method, why didn’t the authors include the updated bounding boxes in its evaluation?
> >
> > **Pose Issue:** Thank you for providing additional examples. Unfortunately, the changes in pose are minimal, and it appears that visual information unrelated to the object's appearance is copied from the source image.
> >
> > Specifically, in Figure 12 (bottom row), while the dog is indeed swimming in the water, its pose remains very similar to the one in the source image. Moreover, the parts of the dog that are cut off in the source image also do not appear in the generated image. The same issue occurs with the dog in the first row: the missing legs in the source image are also absent in the generated image. Finally, it appears that the red toy object, which underwent the most significant pose change, lost some of its source features, such as the shape of its eyes.
> >
> > This figure (along with Figures 8 and 5) highlights my main concern—it seems the method struggles significantly with generalizing beyond the source image. Rather than generating the object in novel poses or filling in the gaps of the source image, it constructs a scene around the object to compensate for these limitations.
> >
> > What I hoped to see in these examples are significant pose changes that require the method to leverage the knowledge contained in the underlying text-to-image model. Some examples from prior work include:
> >
> > * Figure 1 of [1]: A sleeping dog depicted in a pose vastly different from the source images.
> > * Figure 1 of [2]: Depictions of the shoe or the stuffed toy in different poses.
> >
> > [1] DreamBooth: Fine Tuning Text-to-Image Diffusion Models for Subject-Driven Generation
> >
> > [2] AnyDoor: Zero-shot Object-level Image Customization

---

> ### Author Response · Authors · 2024-12-03
> **Response to Reviewer QbKw**
>
> Dear Reviewer QbKw:
>
> Thanks for your further response.
>
> (1) Quantitative comparison:
>
> We have updated the results for AnyDoor using the save bounding box scale normalization method as we previously mentioned, the result is shown as below (**Table A**). The results demonstrate that our method achieves improved CLIP-T, CLIP-I and DINO-I scores, outperforming all baseline personalized text-to-image generation methods and layout-guided text-to-image generation methods in this case.
>
> |                | CLIP-T ↑   | CLIP-I ↑   | DINO-I ↑   |
> |----------------|------------|------------|------------|
> | BLIP-Diffusion | 0.2824     | 0.8894     | 0.7625     |
> | ELITE          | 0.2461     | 0.8936     | 0.7557     |
> | Kosmos-G       | 0.2864     | 0.8452     | 0.6933     |
> | lambda-eclipse | 0.2767     | 0.8973     | 0.7934     |
> | AnyDoor        | 0.2442     | 0.9071     | 0.7932     |
> | GLIGEN         | 0.2898     | 0.8520     | 0.6890     |
> | CustomNet      | 0.2821     | 0.9103     | 0.7587     |
> | **Ours**       | **0.2911** | **0.9169** | **0.7950** |
>
> (2) About the pose:
>
> For the red toy case, we want to clarify that, as shown in row 4 of **Fig. 12**, our method successfully preserves identity details even under significant changes in viewpoint and pose, as directed by the text prompt. Compared to the two other recent text-to-image customization methods, our generated results exhibit eyes that are much more similar to those of the reference subject, making our method the best at maintaining the key features and details.
>
> Actually there isn’t a clear definition for identity preservation in customization task. It is still an open question whether the generated object should tightly follow the appearance of the original reference object or let the model dream and fill the missing parts. In our work, we follow the standard setting of novel view synthesis, where we keep the original appearance of the object as much as possible with an appropriate pose and viewpoint change. For instance, suppose a reference chair has only 3 legs. For our model, we just change the pose that does not compensate for the missing legs. This is reasonable as we accurately maintain the intrinsic property of the object. During our training, we construct our training image pairs where the target image exactly follow the appearance of the input reference image with only pose and viewpoint change. We do not conduct inpainting for the missing parts of the object. This ensures that our model can exactly follow the appearance of the reference object.
>
> As the deadline of revising PDF has passed, we are not able to show more visualization results. However, we would like to emphasize again that our method achieves an excellent balance among identity preservation, text alignment, grounded generation capability, pose adjustment and foreground-background harmonization. Accomplishing all these tasks simultaneously is inherently challenging. Although we cannot guarantee for every task we are perfect, allover our method is significantly better than all the existing methods. Moreover, our method is capable of addressing multiple tasks concurrently, unlike many others methods who only focus on text-to-image customization or layout-guided text-to-image generation. It is unfair to focus solely on one aspect of evaluation while disregarding the substantial advancements we have made in other aspects.
>
> Best,
>
> GroundingBooth Authors

---

### Official Review · Reviewer_PBt1 · 2024-11-03

**Soundness:** 3
**Presentation:** 3
**Contribution:** 3
**Rating:** 6
**Confidence:** 3

**Summary:**

This paper introduces GroundingBooth, a novel framework designed to enhance text-to-image customization by enabling precise spatial control of both subjects and background elements based on textual prompts. While existing models in text-to-image generation maintain subject identity, they often lack control over spatial relationships. GroundingBooth addresses this gap by implementing zero-shot instance-level spatial grounding, enabling precise placement of both foreground subjects and text-defined background elements.
GroundingBooth supports complex tasks such as multi-subject customization, where multiple subjects and background entities are positioned according to input bounding boxes. Experimental results demonstrate its effectiveness in layout alignment, identity preservation, and text-image alignment, outperforming current approaches in controlled image generation.

**Strengths:**

Unlike many existing layout-guided image generation methods that handle only single subjects, GroundingBooth supports multi-subject customization. This versatility broadens its applicability, especially for generating images where complex layouts and multiple subjects are essential.

**Weaknesses:**

1. InstanceDiffusion does not exist in baseline comparisons. Despite its notable relevance with capabilities for free-form language conditions per instance and flexible instance localization methods (single points, scribbles, and bounding boxes), InstanceDiffusion is missing from both our quantitative and qualitative baselines.
2. FID, in contrast to other works dealing with similar tasks, is not suggested in this paper.
3. Qualitative results demonstrating the model's performance on multi-subject generation tasks are notably absent from this paper.

**Questions:**

1. Previous research in layout-guided diffusion has demonstrated limitations in maintaining visual coherence when objects exhibit diverse textures. While these approaches often resulted in disharmonious image generation, our proposed method provides users with the capability to directly select and manipulate subjects. A comparative analysis with InstanceDiffusion would be particularly valuable, especially in terms of texture consistency and user control capabilities.
2. Due to the lack of publicly available code and data, an accurate evaluation is difficult to conduct.
3. It remains unclear why the paper emphasizes its zero-shot capability as a key strength even though the methodology clearly includes training procedures within the paper.([L37-40] & [L247-250])

---

> ### Author Response · Authors · 2024-11-21
> **Response to Reviewer PBt1**
>
> Dear Reviewer PBt1,
>
> Thanks a lot for your insightful feedback and kind advice. We would like to address your concerns one by one.
>
> 1. Comparison with InstanceDiffusion[1]：
>
> We need to first emphasize that InstanceDiffusion is a layout-guided text-to-image generation method. It cannot maintain the identity of the reference object. The qualititive comparison of our method with InstanceDiffusion conditioned on layout is shown in **Fig.7** of the updated PDF.  We test the model’s performance on the non-filtered whole coco validation dataset with 5,000 images, the results are shown below. InstanceDiffusion is a pure grounded text-to-image generation method and is not able to maintain the identity of the reference subjects, which is reflected in the CLIP-I and DINO scores. Compared with InstanceDiffusion, our method shows better results in both text alignment, identity preservation, and layout alignment.
>
> Table A: Comparison with InstanceDiffusion
>
> |  | CLIP-T ↑| CLIP-I ↑| DINO-I ↑| AP50 ↑|
> | --- | --- | --- | --- | --- |
> | InstanceDiffusion | 0.2914 | 0.8391 | 0.7939 | 37.2 |
> | Ours | **0.2968** | **0.9095** | **0.8592** | **38.3** |
>
> 2. FID is not suggested in the paper：
>
> For DreamBench, as there is no ground truth for the reference objects, it is not appropriate to use FID to evaluate the model’s performance. We report the FID score on the coco validation set, the results are shown below. Results show that our method obtains much better FID metrics than layout-guided text-to-image methods.
>
> Table B: Evaluation on FID score
>
> |  | LAMA | LayoutDiffusion | UniControl | GLIGEN | InstanceDiffusion | Ours |
> | --- | --- | --- | --- | --- | --- | --- |
> | FID↓ | 69.50 | 37.90 | 42.22 | 33.14 | 37.57 | **25.63** |
>
> 3. Qualitative results demonstrating the model's performance on multi-subject generation tasks：
>
> You can see qualitative results on multi-subject customization in both **Fig.1** and **Fig. 6**. Please let us know if these aren’t sufficient in a specific way.
>
> Here we add some quantitative evaluation to help demonstrate our model in this setting. Since there are no previous works that evaluate this, we propose a Multi-DINO(M-DINO) and Multi-CLIPI(M-CLIPI) score to evaluate by first computing the DINO/CLIP-I  score between each reference object and the generated image and then calculating the average score.  We test the case for 2 reference objects, where the reference objects and the text descriptions are randomly composited from DreamBench. The results on DreamBench are as follows:
>
> Table C: Evaluation of multi-subject customization
>
> |  | CLIP-T | M-CLIPI | M-DINO |
> | --- | --- | --- | --- |
> | Ours | 29.25 | 0.904 | 0.755 |
>
> These results show that our model maintains text alignment and identity preservation in the multi-subject grounded text-to-image customization task.
>
> 4. A comparative analysis with InstanceDiffusion would be particularly valuable, especially in terms of texture consistency and user control capabilities:
>
> As shown in **Fig.7** of the revised version of the paper, InstanceDiffusion cannot maintain the identity of the reference object. From the table above, we can see that our model get better performance in text-alignment, identity preservation, and layout alignment.
>
> 5. About the publicity of code and data:
>
> The benchmark datasets are all publicly available. Our code will be available upon acceptance.
>
> 6.  Why the paper emphasizes its zero-shot capability as a key strength?
>
> Our zero-shot means that our method does not need test-time finetuning during the inference phase.  As we have summarized in the related work, customization methods mainly include test-time fine-tuning-based methods, which means that users need to finetune the model for every new subject; and zero-shot methods, which means once the model is trained, users do not need to fine-tune the model for every new subject in the inference stage. Under this definition, our method belongs to the zero-shot method. Compared with test-time-finetuning methods, zero-shot methods are more efficient and flexible and can be easily generalized to unseen objects and scenarios.
>
> [1] InstanceDiffusion: Instance-level control for image generation

---

> > ### Author Response · Authors · 2024-11-22
> >
> > Dear Reviewer,
> >
> > We hope our responses have adequately addressed your previous concerns about (1) comparison with InstanceDiffusion, (2) further quantatitive evaluation and (3) definition of zero-shot capability. We look forward to hearing from you and would be happy to address any remaining concerns that you may still have.
> >
> > Thanks,
> >
> > Authors

---

> > > ### Author Response · Authors · 2024-11-24
> > > **Kindly Reminder**
> > >
> > > Dear Reviewer PBt1,
> > >
> > > We sincerely appreciate the time and effort you have dedicated to reviewing our submission. We have submitted our response and would like to follow up to inquire whether our response has sufficiently addressed your concerns.
> > >
> > > Please do not hesitate to let us know if you have any remaining questions or require additional clarification. We are glad to address your further concerns.
> > >
> > > Thank you once again for your valuable insights and guidance.
> > >
> > > Best regards,
> > >
> > > GroundingBooth Authors

---

> ### Comment · Reviewer_PBt1 · 2024-11-26
>
> Thank you for your supportive experiments and justification! You have addressed my concerns questions, so I have increased my score to positive.

---

> > ### Author Response · Authors · 2024-11-27
> >
> > Dear Reviewer PBt1,
> >
> > Thank you very much for recognizing our work and providing valuable feedback that has helped improve the quality of our paper. Your input has been crucial in enhancing our research, and we sincerely appreciate your constructive comments and support.
> >
> > If you have any further questions or suggestions, we would be more than happy to continue the discussion.
> >
> > Best regards,
> >
> > GroundingBooth Authors

---

### Official Review · Reviewer_SAXF · 2024-11-03

**Soundness:** 3
**Presentation:** 2
**Contribution:** 3
**Rating:** 6
**Confidence:** 4

**Summary:**

The paper proposes GroundingBooth, a model for grounded text-to-image customization. It aims to place subjects received in input (marked with bounding-boxes in images) in new backgrounds (described in the prompt), while maintaining the identity and spatial location of the subjects. The authors show GroundingBooth is capable of generating complex requests while preserving the subjects in the input images (e.g., “a [stuffed animal] and a [vase] with [plant] and [vintage lantern] on a quaint balcony”)

GroundingBooth incorporates a new Masked Cross Attention module in each block of the U-Net (Stable Diffusion 1.4’s). In addition to input from the existing Cross Attention layer, the masked layer receives as input DINO-2 features of the subject images received in the input. GroundingBooth is trained this way on a dataset curated from MVImgNet.

Finally, the method is tested and compared to a few existing baselines, using automatic measurements such as CLIPScore and DINO, and a human study.

**Strengths:**

* The paper is well written and presented nicely
* The method improves over the baselines it does test (see first weakness)
* Such model can be useful in many real-life applications

**Weaknesses:**

* The paper does not cover “Break-A-Scene: Extracting Multiple Concepts from a Single Image” by Avrahami et al (2023). In this work, they extract concepts from an image using textual inversion, and use it to embed them in new images. They too work with masks and can even accept them from the user as input. This is especially important since the sentence before last in the abstract states “Our work is the first work to achieve a joint grounding of both subject-driven foreground generation and text-driven background generation”, which makes this imprecise. More importantly, the difference between these projects should be clearly stated. What does this work do that Break-A-Scene does not?

* The use of Fourier embedding should be explained. What makes it suitable to this task?

**Questions:**

* Why does this method use SD-1.4 when there are so many newer / stronger models? Is there some limitation in using them?

---

> ### Author Response · Authors · 2024-11-21
> **Response to Reviewer SAXF**
>
> Dear Reviewer SAXF:
>
> Thanks a lot for your valuable questions. We will address your concerns one by one:
>
> Q1: Comparison with Break-a-scene[1]:
>
> A1: Thanks for pointing this out. The focus of our work is encoder-based customization methods, so this statement has an implicit condition that we mainly compare our methods with existing customization methods without test-time fine-tuning. We have cited this work in the revised version of paper **L114-L115**. There is a large difference between our paper and Break-a-scene:
> 1. Break-a-scene is a test-time-finetuning-based method, while our work is encoder-based work which does not need test-time finetuning.
> 2. In their paper, they only show results about grounding foreground objects, while there is no clear evidence showing that they are able to ground background text entities.  Our method achieves a grounded generation of foreground subjects and background text entities at the same time.
>
> Q2: About Fourier embedding:
>
> For Fourier embedding, it is a common method used in text-to-image generation to encode the position information. Fourier embeddings can encode bounding box coordinates or region-specific positional cues to generate content grounded in specific areas of an image. It is a method of positional encoding, which is suitable to encode the bounding box information.
>
> Q3: About the version of Stable Diffusion:
>
> A3: Our work is mainly designed to evaluate the effectiveness of our proposed pipeline of grounded text-to-image personalization.  Our pipeline can also be easily extended to other diffusion architectures. As we stated in **L537-L538** of the paper, applying it to the state-of-the-art diffusion models will be one of the future works.
>
> Reference:
> [1] Break-A-Scene: Extracting Multiple Concepts from a Single Image

---

> > ### Author Response · Authors · 2024-11-23
> >
> > Dear Reviewer,
> >
> > We hope our responses have adequately addressed your previous concerns about (1) comparison with prior works, (2) details about Fourier embedding and (3) the version of Stable Diffusion. We really look forward to hearing from you and would be happy to discuss and address any remaining concerns that you may still have.
> >
> > Thanks,
> >
> > Authors

---

> > > ### Author Response · Authors · 2024-11-24
> > > **Kindly Reminder**
> > >
> > > Dear Reviewer SAXF,
> > >
> > > We sincerely appreciate the time and effort you have dedicated to reviewing our submission. We have submitted our response and would like to follow up to inquire whether our response has sufficiently addressed your concerns.
> > >
> > > Please do not hesitate to let us know if you have any remaining questions or require additional clarification. We are glad to address your further concerns.
> > >
> > > Thank you once again for your valuable insights and guidance.
> > >
> > > Best regards,
> > >
> > > GroundingBooth Authors

---

> > ### Comment · Reviewer_SAXF · 2024-11-27
> >
> > Thank you for your rebuttal.
> >
> > Q1 / W1: Your revision cites the paper in passing, it does not distinguish your work from theirs. This paper is very relevant to your work, and despite your explanation, I do not see in what way this work introduces new capabilities over those introduced in Break-A-Scene, and I find the statement ("Our work is the first ...") in your abstract to be misleading. Provide examples. There is no reason to believe the examples in the paper are different from those shown in Break-A-Scene.
> >
> > To be clear, like I have originally explicitly stated, there needs to be a paragraph explaining how GroundingBooth is similar to Break-A-Scene as well as how they are different, it cannot be in passing, coupled with four more citations.
> >
> > Q3: If it can be easily extended to new models, it should, as SD-1.4 is somewhat outdated with most people working with transformer-based architectures, like FLUX and SD-3. If there is some other explanation that makes it a non-trivial effort, then please provide it.

---

> ### Author Response · Authors · 2024-11-27
> **Kindly Reminder**
>
> Dear Reviewer SAXF,
>
> We sincerely appreciate the time and effort you have dedicated to reviewing our submission. We have submitted our rebuttal and would like to follow up to inquire whether our responses have sufficiently addressed your concerns.
>
> Please let us know if you have any remaining questions or require additional clarification. We value your feedback and are eager to ensure our work meets the highest standards.
>
> Thank you again for your thoughtful insights and guidance.
>
> Best regards,
>
> GroundingBooth Authors

---

> ### Author Response · Authors · 2024-11-28
> **Response to Reviewer SAXF**
>
> Thanks for your response.
>
> Q1: About comparison with Break-A-Scene:
>
> Thanks again for your suggestions,  we agree that this paper is very relevant to our paper, we will compare with this paper in the final version.
>
> We summarize the similarity and difference between our method and Break-A-Scene as below:
>
> Similarity:
>
> (1) Both Break-A-Scene and our method can achieve grounded generation of foreground reference subjects.
>
> (2) Both Break-A-Scene and our method can achieve multi-subject-driven personalized text-to-image generation.
>
> Differences:
>
> (1) Break-A-Scene is a test-time fine-tuning method, whereas our approach is encoder-based and does not require test-time fine-tuning, resulting in faster and more efficient inference.
>
> (2) Our method achieves joint grounding of subject-driven foreground generation and text-driven background generation. For Break-A-Scene, there is no clear evidence in their paper that it can perform grounded generation for the text-driven background objects.
>
> (3) In Break-A-Scene, there is no conclusive evidence that it can perform grounded generation for multiple subjects given each subject's position and prompt.  They only show results about iterative local editing given the background and the editing region in their Fig.10(d). Similarly, Fig.9 does not demonstrate grounded generation for multiple concepts driven by text entities. Our method is capable of simultaneously performing grounded, customized generation for multiple subjects and multiple text entities.
>
> (4) Break-A-Scene employs a cross-attention loss in the diffusion process to attent concepts to the corresponding regions. Our approach, however, integrates both a grounding module and masked cross-attention layers to achieve joint grounded generation of foreground subjects and text-guided background.
>
> We have already summarized the similarity and difference in the **L132-L139** (marked red) of the revised PDF and also make some revisions on the abstract.
>
> Q3: About SD V1.4:
>
> Thanks again for your suggestions. Actually we have been thinking about using new models in our work.  For fair comparison, currently many methods like GLIGEN[1], BLIP-Diffusion[2], KOSMOS-G[3] that we compared with are based on SD V1.4 or SD V1.5. Meanwhile,  we have found that the official FLUX training code is currently not available for use. While SD-3 has been released, fine-tuning this model demands significantly more computational resources than we currently have available. Therefore based on the factors above, we use SD V1.4 as our base model. We leave transfering to these frameworks as our future works.
>
> References:
>
> [1] Li Y, Liu H, Wu Q, et al. Gligen: Open-set grounded text-to-image generation. CVPR 2023.
>
> [2] Li D, Li J, Hoi S. Blip-diffusion: Pre-trained subject representation for controllable text-to-image generation and editing. NeurIPS 2024.
>
> [3] Pan X, Dong L, Huang S, et al. Kosmos-g: Generating images in context with multimodal large language models. ICLR 2024.

---

### Official Review · Reviewer_gFc1 · 2024-11-04

**Soundness:** 2
**Presentation:** 2
**Contribution:** 2
**Rating:** 6
**Confidence:** 3

**Summary:**

This paper proposes a framework which allows users to customize an image by 1) specifying the position (layout) of the object, and 2) providing a reference image of the object. It supports either single object customization or multi-object customization. They design a grounding module to ground the provided image with text entities. The produced grounding tokens are then later used as the condition in their diffusion model to generate the final image. They conduct experiments on Dreambench and MS-COCO and show that their methods could produce high quality image while preserving the detail of the user-specified (reference) images.

**Strengths:**

1. The visualization results show that the proposed method can effectively preserve the identity of reference image while generating plausible images.
2. The proposed method is able to simultaneously handle multi-object synthesis even with complex layout.

**Weaknesses:**

1. My main concern is that the authors claim that they are able to ground the text entities during generation. While the CLIP-T score of the model indicates that the generated image is less coherent with the text comparing to other baseline methods.
2. While the paper claimed that they can control the spatial relationship between objects. It is difficult to evaluate this argument given the layouts are pre-determined.
3. How are the metrics computed? For example, when computing the CLIP-I score, do you only consider the image similarity between the reference object and the corresponding region in the generated image? If so, how do you extract the corresponding region? More details of how the metrics are computed (CLIP-I, DINO, CLIP-T) could improve the clarity of the paper.

**Questions:**

1. For multi-objects cases, is each box in the layout assigned to an associated object label?
2. In your experiments, are all the layouts pre-determined or only the layout of the reference object is given?

---

> ### Author Response · Authors · 2024-11-21
> **Response to Reviewer gFc1**
>
> Dear Reviewer gFc1,
>
> Thanks a lot for your valuable questions. We will address your concerns one by one.
>
> 1. About the question about the CLIP-T score:
>
> The stable diffusion base model plays a vital role in determining the upper bound of the CLIP-T score. We have already illustrated this point in **L537-L539** of the paper. Although our model is based on SD V1.4, it still shows competitive scores compared with other personalized text-to-image generation methods that use more recent baseline SD models. Our method provides a pipeline for grounded text-to-image customization. It can be transferred to other diffusion architectures and we are sure that it will improve the model’s text-alignment ability. For our task need to maintain the text alignment, identity preservation, and layout alignment at the same time, it is a pretty challenging task. Our model can deal with these aspects at the same time while still maintaining competitive scores compared with both personalized text-to-image generation methods and layout-guided text-to-image generation methods, which demonstrates the effectiveness of our model.
>
> 2. About the spatial relationships between objects:
>
> Our motivation is to manipulate the layout to control the reference object generation. The task is that users can provide or manipulate the layout, and the model will generate visual content with layout alignment. The spatial relationship is determined by the bounding boxes of the objects. That is exactly what we would like to emphasize.
>
> 3. About the details of the evaluation metrics:
>
> For the evaluation metrics on Dreambench, we use the ground-truth mask on the reference image and obtain a reference object without background. The ground-truth mask is provided by state-of-the-art segmentation method SAM as we mentioned in **L160-L161** in the submission. For the generated image, we do not conduct mask manipulation and directly compute the scores between the masked reference image and the generated image. As we need to make a fair comparison with previous methods,  and most of the generated images are object-centered images, the results are similar to masked methods. For evaluation metrics computation, we pass the masked reference image and generated image through the CLIP image encoder respectively, and calculate the cosine similarity between the CLIP image embedding of the masked reference image and the generated image. For the CLIP-T score, we compute it between the CLIP text embedding of the input caption, and the generated image embedding.  For the DINO score, we extract the DINO features of the masked reference image and the generated image and compute the cosine similarity between the two embeddings.
>
> For evaluation metrics on COCO,  since it is a fine-grained generation task and we have the ground truth, we compute the evaluation metrics between the generated image and the ground truth image. We follow the same setting to report the results of our method and all the baseline methods. For evaluation metrics computation, we pass the ground-truth image and generated image through the CLIP image encoder respectively, and calculate the cosine similarity between the CLIP image embedding of the masked reference image and the generated image. For the CLIP-T score, we compute it between the CLIP text embedding of the input caption, and the generated image embedding.  For the DINO score, we extract the DINO features of the ground-truth image and the generated image and compute the cosine similarity between the two embeddings.
>
> 4. For multi-objects cases, is each box in the layout assigned to an associated object label?
>
> For multi-object cases, each box in the layout is assigned to either a reference object, text entity, or both. As we illustrated in **Sec. 3.1, Line213-232**,  if a bbox refers to the reference object, both the text label and the reference object image are used to get the corresponding text and image tokens.  For the boxes where there is no reference object or text entities, we set the input reference object layout to [x1,y1,x2,y2]=[0.0,0.0,0.0,0.0] and reference object token to zero embeddings, or set the grounded text embeddings to zero embeddings, respectively.
>
> 5. In the experiments, are all the layouts pre-determined or only the layout of the reference object is given?
>
> In the quantitative experiments on DreamBench (**Table 1**), to make a fair comparison with other grounded text-to-image customization methods, as there is no ground truth, we set the layout of the reference object to be **the same** as the layout in the reference image. For quantitative experiments on COCO(**Table 2**), we set the layout of the reference object to be **the same** as the layout in the ground-truth image. In visualization, the layouts of the objects are randomly generated. We should emphasize that our model can take bounding boxes of reference objects or background text entities as input.

---

> > ### Author Response · Authors · 2024-11-22
> >
> > Dear Reviewer,
> >
> > We hope our responses have adequately addressed your previous questions about (1) CLIP-T score, (2) spatial relationships, (3) the details of the evaluation metrics, (4) multi-objects cases and (5) detail of layouts. We look forward to hearing from you and would be happy to address any remaining concerns that you may still have.
> >
> > Thanks,
> >
> > Authors

---

> > ### Comment · Reviewer_gFc1 · 2024-11-26
> > **Thanks for the response**
> >
> > Thank you for the detailed response! I still have a couple of follow-up questions:
> >
> > 1. The authors stated that "our model is based on SD V1.4." However, in Table 1, the proposed method’s CLIP-T score is lower than that of SD V1.4. This seems to contradict the paper's main claim that the proposed method effectively grounds text entities during generation. Could you clarify this discrepancy?
> >
> > 2. If the layouts are predetermined, it raises concerns about the technical novelty of the paper. Aligning output images with layouts has already been demonstrated using layout + GAN approaches, not to mention layout + diffusion methods. Could you explain how this work advances beyond these existing techniques?

---

> > > ### Author Response · Authors · 2024-11-27
> > > **Response to Reviewer gFc1**
> > >
> > > Thank you for your response. We will answer the questions one by one.
> > >
> > > 1. About comparison with Stable Diffusion V1.4 :
> > >
> > > From our experiments, we observed that while text grounding is highly effective for generating text-aligned objects, there are instances where certain words are not associated with bounding boxes, resulting in these regions being generated solely through text-to-image mechanisms. In scenarios where the model needs to perform multi-task generation(combining text alignment, identity preservation, and layout alignment), the text-alignment performance on multi-task model  tends to decline compared to single-task models such as the base model SD V1.4, since the model need to take into account multiple tasks.
> > >
> > > This phenomenon can be observed in the results of GLIGEN[1] in **Table 1** and **Table 6** in the revised PDF, which also employs the SD V1.4 base model for layout-guided text-to-image generation. Unlike our approach, GLIGEN has fewer tasks to address, as it does not involve identity preservation. Nonetheless, it faces similar issues, showing degraded performance in areas that are not grounded by bounding boxes. Similar trends are seen in other recent multi-task text-to-image customization methods, such as Lambda-Eclipse[2], KOSMOS-G[3], BLIP-Diffusion[4], as well as grounded text-to-image generation methods like GLIGEN, all of which exhibit a decline in CLIP-T scores compared to their respective base models in multi-task scenarios. The performance of text alignment on the base model can be considered as the upper bound for subsequent multi-task methods.
> > >
> > > 2. About advances beyond these existing techniques:
> > >
> > > We need to emphasis the definition of personalized text-to-image generation:  **Utilizing single or multiple images that contain the same subject, along with text prompt, to generate images that contain that subject as well as match the textual description.**
> > >
> > > The methods you mentioned only layout-guided text-to-image generation methods, and are not designed for personalized text-to-image generation. As such, they can not achieve identity preservation. As explained in **L064-L072**, our grounded text-to-image customization task is not simply about aligning layouts. A key aspect of our task is identity preservation, which is quantitatively evaluated through metrics such as CLIP-I and DINO-I scores. Identity preservation presents a significant challenge in our experiments.
> > >
> > > Our method not only accomplishes grounded text-to-image generation, but also supports reference image input, enabling joint grounding of the foreground reference image along with background text entities. Moreover, the generated reference object is capable of achieving pose changes that harmonize naturally with the background, resulting in a nuanced and coherent scene. This makes our approach more sophisticated than existing layout-guided text-to-image generation methods, as it addresses the more complex problem of maintaining fine-grained identity preservation and interaction between subjects and background.
> > >
> > > [1] Li Y, Liu H, Wu Q, et al. Gligen: Open-set grounded text-to-image generation. CVPR 2023
> > >
> > > [2] Patel M, Jung S, Baral C, et al. lambda-ECLIPSE: Multi-Concept Personalized Text-to-Image Diffusion Models by Leveraging CLIP Latent Space. Arxiv 2024
> > >
> > > [3] Pan X, Dong L, Huang S, et al. Kosmos-g: Generating images in context with multimodal large language models. ICLR 2024
> > >
> > > [4] Li D, Li J, Hoi S. Blip-diffusion: Pre-trained subject representation for controllable text-to-image generation and editing. NeurIPS 2024

---

> > > > ### Author Response · Authors · 2024-11-29
> > > > **Kindly Reminder**
> > > >
> > > > Dear Reviewer gFc1:
> > > >
> > > > We sincerely appreciate the time and effort you have dedicated to reviewing our submission. We have submitted our response and would like to follow up to inquire whether our response has sufficiently addressed your concerns.
> > > >
> > > > Please feel free to let us know if you have any remaining questions or if further clarification is needed. We are still eager to engage in further discussion and address any additional concerns you may have. If you find our response addressed your concern, we would deeply appreciate it if you could consider raising our rating.
> > > >
> > > > Thank you once again for your valuable insights and guidance.
> > > >
> > > > Best regards,
> > > >
> > > > GroundingBooth Authors

---

> > > > > ### Comment · Reviewer_gFc1 · 2024-12-01
> > > > > **Thanks for the response**
> > > > >
> > > > > Thank you for your response. I'm willing to raising the score to 6. However, I would appreciate it if the author could further highlight the technical significance of identity preservation in text-to-image generation in the introduction and provide more clarity on how this is achieved in the method section. The approach to enforcing the preservation constraint is somewhat unclear in the current manuscript.

---

> > > > > > ### Author Response · Authors · 2024-12-01
> > > > > >
> > > > > > Dear Reviewer gFc1:
> > > > > >
> > > > > > Thank you very much for recognizing our work and providing valuable feedback that has helped improve the quality of our paper. Your input has been crucial in enhancing our research, and we sincerely appreciate your constructive comments and support. As you suggested, in the final version, we would highlight the technical significance of identity preservation in text-to-image generation in the introduction and provide more clarity on how this is achieved in the method section.
> > > > > >
> > > > > > If you have any further questions or suggestions, we would be more than happy to continue the discussion.
> > > > > >
> > > > > > Best regards,
> > > > > >
> > > > > > GroundingBooth Authors

---

> ### Author Response · Authors · 2024-11-24
> **Kindly Reminder**
>
> Dear Reviewer gFc1,
>
> We sincerely appreciate the time and effort you have dedicated to reviewing our submission. We have submitted our response and would like to follow up to inquire whether our response has sufficiently addressed your concerns.
>
> Please do not hesitate to let us know if you have any remaining questions or require additional clarification. We are glad to address your further concerns.
>
> Thank you once again for your valuable insights and guidance.
>
> Best regards,
>
> GroundingBooth Authors

---

### Author Response · Authors · 2024-11-21

Thank you to all reviewers for dedicating your time to review our paper, and for providing valuable and insightful feedback. We are thrilled that the reviewers acknowledge our noteworthy generation results and the novel approach of grounded text-to-image customization.

We've updated our paper to include additional ablation studies and image generation results, which you can find in the main content(marked red) and Appendix E, F, G, H and I, respectively. These updates are intended to address your concerns about the mode’s ability to prevent context misplacement, analysis of the number of reference objects, the pose change of the generation, the ablation study about grounding circumstance, and the interactions between reference objects respectively.

We look forward to hearing from you. We have carefully addressed the main concerns and provided detailed responses to each reviewer. We hope you will find the responses satisfactory. If you have any further questions or concerns, please do not hesitate to let us know. We are eager to address them promptly before the discussion deadline.

---

### Meta-Review · Area_Chair_PXrK · 2024-12-18

**Metareview:**

This paper proposes a grounded text-to-image generation framework incorporating reference objects and bounding-box constraints. While  backed by comprehensive experiments, in private discussion period, the reviewers find the contributions incremental. The core techniques, such as masked cross-attention and gated self-attention, have been explored in prior works, such as "Training-Free Layout Control with Cross-Attention Guidance" and "Grounded Text-to-Image Synthesis with Attention Refocusing". object poses remain identical to the reference images, contradicting claims of flexibility. The results do not convincingly outperform existing methods, and the novelty is limited mostly a combination of known ideas without substantial new insights. the reviewers remain unconvinced of the work's originality.

**Additional Comments On Reviewer Discussion:**

For metareview, low-quality reviews are carefully considered during the decision process. Although I was unable to guide more engagement from some reviewers during the discussion period, I placed very low weight on feedback from those who did not actively participate for reviewer discussion.
The rebuttal and extra experiments did not resolve concerns about insufficient novelty and limited improvements over prior works. The reviewers who discussed during the rebuttal period continued to question the originality of this work, pose adaptation capabilities, and practical advantages. Even considering the lack of response from reviewers, the final decision stands as rejection.

---

### Decision · Program_Chairs · 2025-01-22

Reject